# MASK-BASED MODELING FOR NEURAL RADIANCE FIELDS

**Ganlin Yang** [1]  **Guoqiang Wei** [2]  **Zhizheng Zhang** [3]*  **Yan Lu** [3]  **Dong Liu** [1]*
[1] University of Science and Technology of China [2] ByteDance Research [3] Microsoft Research Asia
ygl666@mail.ustc.edu.cn  weiguoqiang.9@bytedance.com
{zhizzhang,yanlu}@microsoft.com  dongeliu@ustc.edu.cn

## ABSTRACT

Most Neural Radiance Fields (NeRFs) exhibit limited generalization capabilities, which restrict their applicability in representing multiple scenes using a single model. To address this problem, existing generalizable NeRF methods simply condition the model on image features. These methods still struggle to learn precise global representations over diverse scenes since they lack an effective mechanism for interacting among different points and views. In this work, we unveil that 3D implicit representation learning can be significantly improved by mask-based modeling. Specifically, we propose **m**asked **r**ay and **v**iew **m**odeling for generalizable **NeRF** (**MRVM-NeRF**), which is a self-supervised pretraining target to predict complete scene representations from partially masked features along each ray. With this pretraining target, MRVM-NeRF enables better use of correlations across different points and views as the geometry priors, which thereby strengthens the capability of capturing intricate details within the scenes and boosts the generalization capability across different scenes. Extensive experiments demonstrate the effectiveness of our proposed MRVM-NeRF on both synthetic and real-world datasets, qualitatively and quantitatively. Besides, we also conduct experiments to show the compatibility of our proposed method with various backbones and its superiority under few-shot cases. Our codes are available at https://github.com/Ganlin-Yang/MRVM-NeRF.

## 1 INTRODUCTION

Neural Radiance Field (NeRF) (Mildenhall et al., 2021) has emerged as a powerful tool for 3D scene reconstruction (Sun et al., 2022; Yu et al., 2021a; Fridovich-Keil et al., 2022) and generation (Niemeyer & Geiger, 2021; Lin et al., 2023; Poole et al., 2022). Though most NeRF-based methods can render striking visual results, they are still restricted to a particular static scene, limiting their application in a wide range. Recent works study *Generalizable NeRF* (Yu et al., 2021b; Wang et al., 2021; 2022b; Reizenstein et al., 2021; Liu et al., 2022) to model various scenes with a single model, which can be directly applied to an unseen scene during inference.

Most of existing methods for generalizable NeRF sample image features from several visible reference views as the conditions for learning scene representations. However, the correlations among the sampled features are not well exploited before. Previous masked modeling tasks, including masked language modeling (MLM) (Devlin et al., 2018) in natural language processing and masked image modeling (MIM) (Bao et al., 2021; Devlin et al., 2018; Xie et al., 2022; He et al., 2022) in computer vision, exploiting such correlations among input signals by a *mask-then-predict* task: masking out a proportion of inputs and trying to predict the missing information from the remaining ones. In this way, a high-level global representation could be learned, which is beneficial for downstream tasks. As for NeRFs, we find that the high-level global information learned through mask-based pretraining, which we call the *3D scene prior knowledge*, is also extremely useful for generalizable Neural Radiance Field. When applying for a novel scene, such a prior knowledge comes to use for reconstructing a high-quality new scene from limited reference views.

---

*Corresponding authors: Z. Zhang and D. Liu

To this end, we propose an innovative *masked ray and view modeling (MRVM)* tailored for NeRF, considering that there are correlations among the sampled points along rays and across the reference views naturally. Specially, we introduce a pretraining objective to predict the complete scene representations from the ones being partially masked along rays and across views, aiming to encourage the inner interactions at the two levels. In view of the nature that NeRFs are implicit representations, and motivated by Grill et al. (2020); Yu et al. (2022), we conduct our proposed predictive pretraining in the latent space and optimize it together with NeRF's original rendering task. After pretraining the generalizable NeRF model with our proposed MRVM, the model is further finetuned either across various scenes or on a specific scene. Such a simple yet efficient masked modeling design is actually a model-agnostic method in the sense that it can be widely applicable to various generalizable NeRF models.

To demonstrate the effectiveness and wide applicability of our proposed MRVM, we conduct extensive experiments both on commonly used large-scale synthetic datasets and more challenging real-world realistic datasets, based on both MLP-based and transformer-based network architectures. Quantitative and qualitative experimental results show that our proposed masked ray and view modeling significantly improves the generalizability of NeRF by rendering more precise geometric structures and richer texture details. Our contributions can be summarized as follows:

- We find 3D implicit representation learning can be significantly improved by mask-based modeling as MLM and MIM, when the inner correlations of 3D scene representations are harnessed in the right manner.
- We present a simple yet efficient self-supervised pretraining objective for generalizable NeRF, termed as **MRVM-NeRF**. To our best knowledge, it is the first attempt to incorporate mask-based pretraining into the NeRF field.
- We conduct extensive experiments over various synthetic and real-world datasets based on different backbones. The results demonstrate the effectiveness and the general applicability of our masked ray and view modeling.

## 2 RELATED WORK

### 2.1 NEURAL RADIANCE FIELDS

**Generalizable NeRF** Vanilla Neural Radiance Field (NeRF) introduced by Mildenhall et al. (2021) requires per-scene optimization which can be time-consuming and computationally expensive. To tackle with the generalization problem across multiple scenes, the network requires an additional *condition* to differentiate them. Several works (Jang & Agapito, 2021; Noguchi et al., 2021; Liu et al., 2021) use a global latent code to represent the scene's identity, while more of the others (Yu et al., 2021b; Wang et al., 2021; Liu et al., 2022; Zhang et al., 2022) extract a pixel-aligned feature map to be unprojected into 3D space. Generalizable NeRFs reconstruct the NeRF model on the fly and can synthesize arbitrary views of a novel scene with a single forward pass.

**Backbones** Several earlier classical NeRF works (Mildenhall et al., 2021; Barron et al., 2021; Yu et al., 2021b; Liu et al., 2022) adopt Multiple-Layer Perception (MLP) as the backbone for scene reconstruction. Recently inspired by great success of Transformer (Vaswani et al., 2017) in computer vision area (Dosovitskiy et al., 2020), there have also been some attempts (Reizenstein et al., 2021; Wang et al., 2022a;b) to incorporate attention mechanisms into NeRF model. We evaluate the efficacy of our mask-based pretraining strategy on one representative work for each backbone.

### 2.2 MASKED MODELING FOR PRETRAINING

Mask-based modeling has been widely used for pretraining in various research domains. In Natural Language Processing, Masked Language Modeling (MLM) is employed to pretrain BERT (Devlin et al., 2018) and its autoregressive variants (Radford et al., 2018; 2019; Brown et al., 2020). In Computer Vision, Masked Image Modeling (MIM) (He et al., 2022; Bao et al., 2021; Xie et al., 2022; Baevski et al., 2022) has also gained significant popularity for self-supervised representation learning. Different from the aforementioned works, we perform *masking* and *predicting* operations both in the latent feature space drawing inspirations from Grill et al. (2020); Yu et al. (2022), which better coordinates 3D implicit representation learning for NeRF.

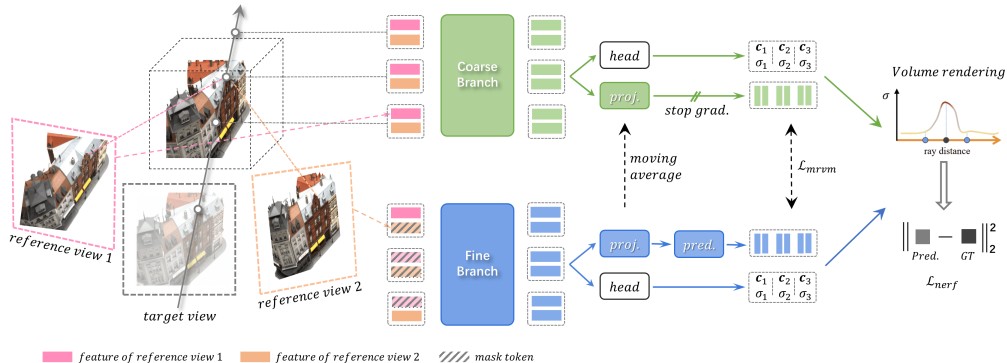

Figure 1: **Overview of our proposed MRVM-NeRF.** To render an image from a target view, rays are cast into 3D space, and a series of points are sampled along each ray. These points are projected onto reference image planes to obtain pixel-aligned image features. We employ a coarse-to-fine sampling strategy and mask a portion of feature tokens input into the fine branch. The coarse and fine branches function as the target and online networks, respectively. Our mask-based pretraining objective $\mathcal{L}_{mrvm}$ aims to predict the corresponding latent representations of the target branch from the online ones within the latent space.

## 3 METHOD

We first briefly introduce the general framework for Generalizable Neural Radiance Field and analyze the benefits of incorporating mask-based pretraining strategy in Section 3.1. We then elaborate on the detailed procedure for mask-based pretraining, referred as masked ray and view modeling (MRVM), in Section 3.2. The pretraining objectives and implementation details are presented in Section 3.3 and Section 3.4 respectively.

### 3.1 GENERALIZABLE NEURAL RADIANCE FIELDS

Generalizable neural radiance field aims to share a single neural network across multiple distinct scenes, which often involves a cross-scene pretraining stage followed by a per-scene finetuning stage. It often conditions the Neural Radiance Field on image features aggregated from several reference views. Supposing $S$ reference views $\{\mathbf{I}^1, \mathbf{I}^2, \ldots, \mathbf{I}^S\}$ are available, pixel-aligned feature maps $\{\mathbf{F}^1, \mathbf{F}^2, \ldots, \mathbf{F}^S\}$ can be extracted using 2D CNNs. To synthesize an image at a target viewpoint, several rays are cast into the scene, $N$ points $\{\mathbf{p}_1, \mathbf{p}_2, \ldots, \mathbf{p}_N\}$ are then sampled along each ray. For each point $\mathbf{p}_i$, its corresponding multi-view RGB components $\{\mathbf{c}_i^1, \mathbf{c}_i^2, \ldots, \mathbf{c}_i^S\}$ and feature components $\{\mathbf{f}_i^1, \mathbf{f}_i^2, \ldots, \mathbf{f}_i^S\}$ can be simply obtained by projecting $\mathbf{p}_i$ onto $S$ reference image planes and sampling from $\mathbf{I}^{1\sim S}$ and $\mathbf{F}^{1\sim S}$. For $j \in [1, S]$, $\mathbf{f}_i^j$ and $\mathbf{c}_i^j$ are often merged and projected to a latent embedding $\mathbf{h}_i^j$. $\mathbf{h}_i^j$, seen as the geometry and texture information acquired from reference view $j$ for point $i$, passes through several blocks of neural network modules for scene-specific information delivery and fusion. The network module can be either MLP or Transformer architecture. In this way, $\mathbf{h}_i^j$ is mapped to the processed latent representation $\mathbf{z}_i^j$. $\{\mathbf{z}_i^j\}_{j=1}^S$ are then pooled among $S$ reference views into the global view-invariant latent feature $\mathbf{z}_i$, which is finally decoded into volume density $\sigma_i$ and color $\mathbf{c}_i$ for ray-marching (Mildenhall et al., 2021).

Although the above-mentioned generalizable NeRF framework has made great success, it uses reconstruction loss only to supervise the learning of the mapping $\mathbf{h}_i^j \to \mathbf{z}_i^j$ from end to end, which is at the core of NeRF's reconstruction. We argue that such a learning scheme lacks an explicit inductive bias to leverage information from other $N-1$ points on the ray and other $S-1$ reference views. Prior works on masked modeling have revealed that the *mask-then-predict* self-supervised task can encourage strong interactions between different input signals. Motivated by this, we propose a mask-based pretraining strategy tailored for NeRF, dubbed **masked ray and view modeling**, to better facilitate the 3D implicit representation learning. The learned *3D scene prior knowledge* encapsulates the correlations among point-to-point and across view-to-view, endowing the model with

better capacity to effectively generalize to novel scenes with limited observations. We'll elucidate the mask-based pretraining strategy in detail in the following.

## 3.2 Masked Ray and View Modeling

We adopt the hierarchical sampling procedure like most NeRF works (Mildenhall et al., 2021; Yu et al., 2021b; Liu et al., 2022). At the coarse stage, we first use stratified sampling within a depth range along the ray, forward the coarse-branch neural network to get the processed latent representation $\mathbf{z}_i^c$ and $\sigma_i^c$, $\mathbf{c}_i^c$ as we described in Section 3.1. At the fine stage, additional points are sampled towards the relevant parts of the surface using importance sampling (Mildenhall et al., 2021). These points, together with those sampled at coarse stage, are processed by the fine-branch neural network, producing $\mathbf{z}_i^f$ and $\sigma_i^f$, $\mathbf{c}_i^f$. We apply the masking operation to all the points processed at fine stage, and further supervise the mask-based pretraining task in a projected feature space apart from the 2D pixel space.

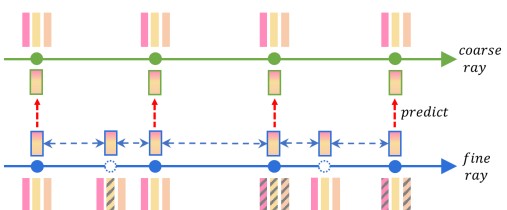

Figure 2: **Illustration of masking operation.** The striped rectangles denote the masked features which are *randomly* selected along the ray. The solid circles represent the points sampled at coarse stage and the hollow ones correspond to extra points sampled at fine stage. The rectangles with solid boxes are processed global view-invariant features by coarse and fine stage, and our MRVM task aims to align them in the same feature space.

We denote the set of points on a single ray at coarse stage as $\mathbf{P}^c$ and fine stage as $\mathbf{P}^f$, while the former is a subset of the latter:

$$\mathbf{P}^c = \{\mathbf{p}_1^c, \mathbf{p}_2^c, \dots, \mathbf{p}_{N_c}^c\}, \tag{1}$$

$$\begin{aligned}\mathbf{P}^f &= \{\mathbf{p}_1^f, \mathbf{p}_2^f, \dots, \mathbf{p}_{N_f}^f\} \\ &= \mathbf{P}^c \cup \{\mathbf{p}_{N_c+1}^f, \mathbf{p}_{N_c+2}^f, \dots, \mathbf{p}_{N_f}^f\},\end{aligned} \tag{2}$$

To facilitate the pretraining of generalizable NeRF, we propose to employ *random masking operation* at two levels, which is illustrated in Figure 2. Specifically, we first perform random masking at the ray-level to enhance the information interaction along each ray, where we randomly select a set of candidate points to be masked out from $\mathbf{P}^f$ according to a preset mask ratio $\eta$. To promote the message-passing across different reference views, we further employ masking at the view-level. For each selected masked point $\mathbf{p}_i^f$, we randomly mask out $1 \sim S$ feature tokens $\{\mathbf{h}_i^j\}_{j=1}^S$ acquired from $S$ reference views.

Similar to Xie et al. (2022), we perform masking simply by replacing the corresponding masked feature token $\mathbf{h}_i^j$ with a shared learnable mask token. In this way, along a specific ray, we randomly discard partial information at certain depths as well as from certain reference views, in accordance with our name **masked ray and view modeling** — masking is executed along cast rays and across reference views, which aligns more closely with the fundamental nature of NeRF.

Advancing beyond previous generalizable NeRFs which solely rely on the pixel-level rendering loss, we aim to further regularize the pretraining process by incorporating constraints within the latent space. Motivated by BYOL (Grill et al., 2020) and several contrastive learning approaches, we designate the unmasked coarse branch as target branch and the masked fine branch as online branch. Our pretraining objective is to align the latent representations associated with the identically sampled points, yet processed through two branches individually. As illustrated in Figure 1, $\mathbf{z}_i^c$ and $\mathbf{z}_i^f$ are further projected to another latent space for feature alignment, which can be formulated as:

$$\bar{\mathbf{z}}_i^c = Proj^c(\Theta, \mathbf{z}_i^c), \tag{3}$$

$$\bar{\mathbf{z}}_i^f = Pred^f(\phi, Proj^f(\theta, \mathbf{z}_i^f)), \tag{4}$$

where $\Theta$, $\phi$ and $\theta$ are corresponding network parameters. The parameters of coarse-projector $\Theta$ are updated by moving average from the fine-projector $\theta$:

$$\Theta \leftarrow \tau\Theta + (1-\tau)\theta, \tag{5}$$

where $\tau \in [0, 1]$ is the moving average decay rate. The MRVM pretraining objective is defined as the feature prediction task in $\overline{\mathbf{z}}$ space:

$$
\begin{aligned}
\mathcal{L}_{mrvm} &= \frac{1}{N_c} \sum_{i=1}^{N_c} \left\| \frac{\overline{\mathbf{z}}_i^f}{\|\overline{\mathbf{z}}_i^f\|_2} - \frac{\overline{\mathbf{z}}_i^c}{\|\overline{\mathbf{z}}_i^c\|_2} \right\|_2^2 \\
&= \frac{1}{N_c} \sum_{i=1}^{N_c} (2 - 2 \frac{\overline{\mathbf{z}}_i^f}{\|\overline{\mathbf{z}}_i^f\|_2} \frac{\overline{\mathbf{z}}_i^c}{\|\overline{\mathbf{z}}_i^c\|_2}),
\end{aligned}
\tag{6}
$$

Note that the constraint is only applied to the points appeared both at coarse and fine stages, *i.e.*, the points in set $\mathbf{P}^c$.

**Discussion** While the mask-based pretraining strategy, as analyzed before, is expected to assist generalizable NeRFs in learning useful 3D scene prior knowledge, there are many mask-based pretraining options. ***Firstly***, directly masking a certain percentage of pixels in reference images, as done in MIM, does not guarantee that each ray sampled during pretraining will be operated masking, which hampers the pretraining efficiency. This is due to the fact that the image features $\mathbf{f}_i^j$ are collected along the epipolar lines on reference image planes, not all of these epipolar lines will pass through the masked pixel regions. ***Secondly***, masking is applied to feature tokens input into the fine branch, because the rendering results of this branch are used for evaluation. Our goal is to enhance the fine-branch's generalization capacity when encountering a novel scene, which is endowed by our mask-learned prior knowledge. Since the coarse branch plays a key role in guiding re-sampling near the surface manifold, it is undesirable to downgrade its accuracy by masking out a portion of its inputs. ***Finally***, the latent representations output from unmasked coarse branch serve as the prediction target, not only by the aspiration for a more streamlined architecture devoid of redundant modules, but also from the inspiration that each of the two branches is dedicated to learning a distinct scale knowledge of the scene, as claimed in MipNeRF (Barron et al., 2021). Consequently, this design choice enables the fine branch neural network to receive a different-scale scene information distilled from the coarse branch. The ablation studies presented in Section 4.3 support our analysis.

### 3.3 TRAINING OBJECTIVES

To help the NeRF model learn better 3D implicit representations during pretraining stage, we also incorporate the conventional NeRF's volume rendering task, and the aforementioned mask-based prediction task in Section 3.2 acts as an auxiliary task to be optimized jointly.

During training, as long as we get the generated color $\mathbf{c}$ and its corresponding density $\sigma$ as described in Section 3.1, we use the classical volume rendering equation (Kajiya & Von Herzen, 1984) to predict the rendering results:

$$
T_i = exp(-\sum_{k=1}^{i-1} \sigma_k \delta_k),
\tag{7}
$$

$$
\mathbf{C}(\mathbf{r}) = \sum_{i=1}^{N} T_i (1 - exp(-\sigma_i \delta_i)) \mathbf{c}_i,
\tag{8}
$$

The rendering loss $\mathcal{L}_{nerf}$ is formulated as:

$$
\mathcal{L}_{nerf} = \|\mathbf{C}^*(r) - \mathbf{C}^c(r)\|_2^2 + \|\mathbf{C}^*(r) - \mathbf{C}^f(r)\|_2^2,
\tag{9}
$$

where $\mathbf{C}^c(r)$ and $\mathbf{C}^f(r)$ are pixel values rendered by coarse and fine branch respectively, and $\mathbf{C}^*(r)$ is the ground truth. The overall pretraining loss is:

$$
\mathcal{L}_{total} = \sum_{\mathbf{r} \in \mathcal{R}} (\mathcal{L}_{nerf} + \lambda \mathcal{L}_{mrvm}),
\tag{10}
$$

where $\lambda$ is set to balance different loss terms.

After pretraining, we perform finetuning as most of the masked modeling works do. The projector and predictor are discarded and no masking operation is performed, only the rendering loss $\mathcal{L}_{nerf}$ is used to update the model until convergence.

Table 1: Quantitative results for category-agnostic **ShapeNet-all** and **ShapeNet-unseen** settings. Detailed breakdown results by categories could be found in Appendix. Best in **bold**.

| Method | ShapeNet-all | | | ShapeNet-unseen | | |
|---|---|---|---|---|---|---|
| | PSNR↑ | SSIM↑ | LPIPS↓ | PSNR↑ | SSIM↑ | LPIPS↓ |
| SRN | 23.28 | 0.849 | 0.139 | 18.71 | 0.684 | 0.280 |
| PixelNeRF | 26.80 | 0.910 | 0.108 | 22.71 | 0.825 | 0.182 |
| FE-NVS | 27.08 | 0.920 | 0.082 | 21.90 | 0.825 | 0.150 |
| SRT | 27.87 | 0.912 | 0.066 | — | — | — |
| VisionNeRF | 28.76 | 0.933 | 0.065 | — | — | — |
| NeRFormer | 27.58 | 0.920 | 0.091 | 22.54 | 0.826 | 0.159 |
| NeRFormer+MRVM | **29.25** | **0.942** | **0.060** | **24.08** | **0.849** | **0.117** |

Table 2: Quantitative results for category-specific **ShapeNet-chair** and **ShapeNet-car** settings, with 1 or 2 reference view(s). Best in **bold**.

| Method | Chair 1-view | | | Chair 2-views | | | Car 1-view | | | Car 2-views | | |
|---|---|---|---|---|---|---|---|---|---|---|---|---|
| | PSNR↑ | SSIM↑ | LPIPS↓ | PSNR↑ | SSIM↑ | LPIPS↓ | PSNR↑ | SSIM↑ | LPIPS↓ | PSNR↑ | SSIM↑ | LPIPS↓ |
| SRN | 22.89 | 0.89 | — | 24.48 | 0.92 | — | 22.25 | 0.89 | — | 24.84 | 0.92 | — |
| FE-NVS | 23.21 | 0.92 | — | 25.25 | 0.94 | — | 22.83 | 0.91 | — | 24.64 | 0.93 | — |
| PixelNeRF | 23.72 | 0.91 | 0.128 | 26.20 | 0.94 | 0.080 | 23.17 | 0.90 | 0.146 | 25.66 | **0.94** | 0.092 |
| CodeNeRF | 23.66 | 0.90 | — | 25.63 | 0.91 | — | 23.80 | 0.91 | — | 25.71 | 0.93 | — |
| VisionNeRF | 24.48 | **0.93** | 0.077 | — | — | — | 22.88 | 0.91 | **0.084** | — | — | — |
| NeRFormer | 23.56 | 0.92 | 0.107 | 25.79 | 0.94 | 0.078 | 22.98 | 0.91 | 0.115 | 25.12 | 0.93 | 0.088 |
| NeRFormer+MRVM | **24.65** | **0.93** | **0.076** | **26.87** | **0.95** | **0.058** | **24.10** | **0.92** | **0.084** | **26.20** | **0.94** | **0.067** |

### 3.4 IMPLEMENTATION DETAILS

To better demonstrate the wide applicability of our proposed mask-based pretraining strategy, we conduct experiments on both MLP-based and transformer-based backbones. Specifically, we adopt NeuRay (Liu et al., 2022) as the MLP-based network, and utilize NeRFormer (Reizenstein et al., 2021) as the transformer-based model. The additional projector and predictor $\Theta$, $\phi$ and $\theta$ are all simple two-layer MLPs. We sample 64 points along each ray at coarse stage, and extra 32 points at fine stage. The moving average decay rate $\tau$ in Equation 5 is set to 0.99, the default mask ratio $\eta$ is set to 50% and the coefficient $\lambda$ for loss term $\mathcal{L}_{mrvm}$ is set to 0.1 during mask pretraining stage unless otherwise stated. Due to the page limits, please refer to the Appendix for more details.

## 4 EXPERIMENTS

To validate the effectiveness of our proposed mask-based pretraining strategy, we conduct a series of experiments under various circumstances. Specifically, we adopt transformer-based backbone under synthetic NMR ShapeNet dataset (Kato et al., 2018), which is introduced in Section 4.1. We also employ MLP-based backbone under realistic complex scenes, with NeRF Synthetic (Niemeyer et al., 2020), DTU (Jensen et al., 2014) and LLFF (Mildenhall et al., 2019) as the three evaluation datasets, presented in Section 4.2. We further conduct a detailed ablation study on 1) mask-based pretraining options, 2) mask ratios as well as 3) few-shot cases in Section 4.3.

**Baselines** We take NeRFormer (Reizenstein et al., 2021) and NeuRay (Liu et al., 2022) as transformer-based and MLP-based baselines respectively. We denote the baselines without any mask-based pretraining as *NeRFormer* and *NeuRay*. Accordingly, the model with MRVM pretraining followed by finetuning is referred as *NeRFormer+MRVM* and *NeuRay+MRVM*. We use PSNR, SSIM (Wang et al., 2004) and LPIPS (Zhang et al., 2018) metrics for evaluation.

### 4.1 EFFECTIVENESS ON SYNTHETIC DATASETS

**Settings** NMR ShapeNet (Kato et al., 2018) is a large-scale synthetic 3D dataset, containing 13 categories of objects. Following the common practices introduced by PixelNeRF (Yu et al., 2021b), we conduct experiments under three settings. 1) In category-agnostic **ShapeNet-all** setting, a single model is trained across all the 13 categories and evaluated over all the 13 categories as well. 2) In category-agnostic **ShapeNet-unseen** setting, the model is trained on *airplane, car and chair* classes

while evaluated on the other 10 categories unseen during training. 3) In category-specific **ShapeNet-chair** and **ShapeNet-car** setting, two models are trained and evaluated particularly on 6591 chairs and 3514 cars respectively, which are subsets of the NMR ShapeNet dataset. For all these settings, we perform masked ray and view modeling simultaneously as we train the generalizable NeRF model across multiple scenes, and evaluate on testing scenes after finetuning without MRVM.

**Results on the category-agnostic setting**    Table 1 shows the quantitative results under category-agnostic **ShapeNet-all** and **ShapeNet-unseen** settings. Under the two settings, each object has 24 fixed viewpoints, with 1 view randomly selected as the reference view while the remaining 23 views used for evaluation. We compare our NeRFormer+MRVM with several dominant generalizable NeRF methods such as SRN (Sitzmann et al., 2019), PixelNeRF (Yu et al., 2021b), FE-NVS (Guo et al., 2022), SRT (Sajjadi et al., 2022) and VisionNeRF (Lin et al., 2022). It can be seen that our baseline NeRFormer has already achieved comparable results with other baseline models. When incorporating mask-based pretraining scheme, its performance is further improved by a large margin in PSNR, SSIM and LPIPS. It demonstrates that the 3D scene prior knowledge learned through our proposed masked ray and view modeling significantly improves the model's generalizability when applying on new scenes.

**Results on the category-specific setting** As for category-specific **ShapeNet-chair** and **ShapeNet-car** settings, during training we randomly provide 1 or 2 reference view(s) for the network with 50 views around per object. During testing, we fix 1 or 2 view(s) as reference(s) and perform evaluation on the rest of views. The experimental results are shown in Table 2. SRN (Sitzmann et al., 2019), FE-NVS (Guo et al., 2022), PixelNeRF (Yu et al., 2021b), Co-deNeRF (Jang & Agapito, 2021) and Vision-NeRF (Lin et al., 2022) are taken as baselines. The enhanced NeRFormer pretrained by our MRVM, *i.e.*, NeRFormer+MRVM, achieves better results than previous methods in both 1-view and 2-view scenarios.

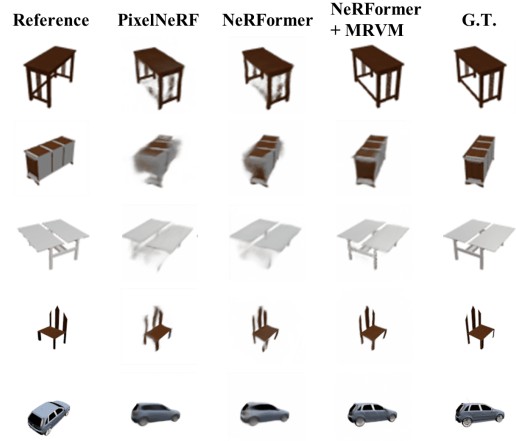

Figure 3: Visualizations of **ShapeNet-all** (row 1-2), **ShapeNet-unseen** (row 3), **ShapeNet-chair** (row 4) and **ShapeNet-car** (row 5) settings. Our MRVM helps render novel views with more plausible structures, finer details and less artifacts.

**Visualizations**    Visual comparisons under the above-mentioned three settings are shown in Figure 3. During pretraining, the MRVM-NeRF model is encouraged to predict the masked information from the rest of available ones, which drives the model to capture the relationship between sampled points and across reference views. At inference, for a novel scene, only partial information is accessible due to the limited reference views, so the mask-learned prior knowledge comes in handy for predicting the implicit representations of unseen parts. Therefore the rendering results have richer details and more precise structures compared to the baselines rendered with blurs and artifacts. More visual results could be found in the Appendix.

## 4.2 Effectiveness on realistic datasets

**Settings**    To further demonstrate that our proposed MRVM is compatible with different NeRF architectures and is applicable beyond simple synthetic datasets, we adopt MLP-based NeuRay (Liu et al., 2022) as the baseline to evaluate on more challenging realistic scenes. Following its protocol, we first pretrain a generalizable NeRF across five datasets: Google Scanned Object dataset (Downs et al., 2022), three forward-facing datasets (Mildenhall et al., 2019; Flynn et al., 2019; Zhou et al., 2018) as well as DTU dataset (Jensen et al., 2014) except for the testing scenes. The masked ray and view modeling is incorporated as an auxiliary task when cross-scene pretraining. We use NeRF Synthetic (Niemeyer et al., 2020), DTU (Jensen et al., 2014) and LLFF (Mildenhall et al., 2019) as evaluation sets following the train-test split manner of NeuRay (Liu et al., 2022). Afterwards, we

Table 3: Quantitative results on NeRF Synthetic, DTU and LLFF datasets. Our proposed MRVM proves to be beneficial for both **cross-scene generalization** and **per-scene finetuning** settings. Best in **bold**.

| | Method | Synthetic Object NeRF | | | Real Object DTU | | | Real Forward-facing LLFF | | |
|---|---|---|---|---|---|---|---|---|---|---|
| | | PSNR↑ | SSIM↑ | LPIPS↓ | PSNR↑ | SSIM↑ | LPIPS↓ | PSNR↑ | SSIM↑ | LPIPS↓ |
| Cross-scene generalization | PixelNeRF | 22.65 | 0.808 | 0.202 | 19.40 | 0.463 | 0.447 | 18.66 | 0.588 | 0.463 |
| | MVSNeRF | 25.15 | 0.853 | 0.159 | 23.83 | 0.723 | 0.286 | 21.18 | 0.691 | 0.301 |
| | IBRNet | 26.73 | 0.908 | 0.101 | 25.76 | 0.861 | 0.173 | 25.17 | 0.813 | 0.200 |
| | NeuRay | 28.29 | 0.927 | 0.080 | 26.47 | 0.875 | 0.158 | 25.35 | 0.818 | 0.198 |
| | NeuRay+MRVM | **29.29** | **0.930** | **0.077** | **29.48** | **0.926** | **0.108** | **26.91** | **0.861** | **0.169** |
| Per-scene finetuning | MVSNeRF | 27.21 | 0.888 | 0.162 | 25.41 | 0.767 | 0.275 | 23.54 | 0.733 | 0.317 |
| | NeRF | 31.01 | 0.947 | 0.081 | 28.11 | 0.860 | 0.207 | 26.74 | 0.840 | 0.178 |
| | IBRNet | 30.05 | 0.935 | 0.066 | 29.17 | 0.908 | 0.128 | 26.87 | 0.848 | 0.175 |
| | NeuRay | 32.35 | 0.960 | 0.048 | 29.79 | 0.928 | 0.107 | 27.06 | 0.850 | 0.172 |
| | NeuRay+MRVM | **33.09** | **0.965** | **0.035** | **31.98** | **0.943** | **0.091** | **28.37** | **0.881** | **0.157** |

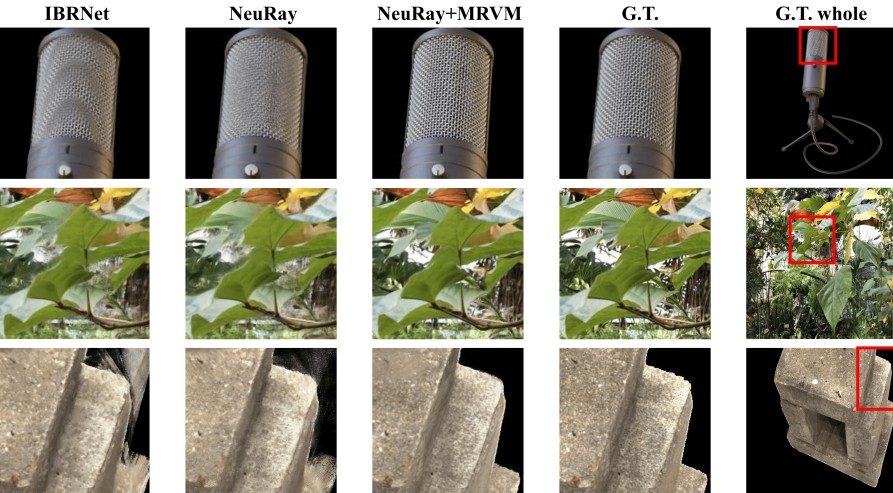

Figure 4: Visualizations on NeRF Synthetic (first row), LLFF (middle row) and DTU (last row) datasets. Masked ray and view modeling aids in rendering images with enhanced texture details, reduced blurring and fewer artifacts.

finetune the generalizable NeRF model without masking operation either across five training sets, dubbed as **cross-scene generalization** setting, or target on a specific scene in the three evaluation sets, denoted as **per-scene finetuning** setting.

**Results** The experimental results can be found in Table 3. We compare our NeuRay+MRVM with several well-known baselines including NeRF (Mildenhall et al., 2021), PixelNeRF (Yu et al., 2021b), MVSNeRF (Chen et al., 2021), IBRNet (Wang et al., 2021) and NeuRay (Liu et al., 2022). The prior knowledge acquired through mask-based pretraining substantially enhances the model's generalization ability when applied to new scenes in **cross-scene generalization** setting (the first large row in Table 3). Furthermore, the prior knowledge is still influential after executing **per-scene finetuning** (the last large row in Table 3). We show the visual comparisons under **per-scene finetuning** setting in Figure 4. More rendering results are placed in the Appendix. The model pretrained by our MRVM delivers better visual effects obviously. It is worth noted that the training and evaluation sets encompass a wide variety of scenes, ranging from single object-centric scenes to more complex forward-facing indoor and outdoor scenes. This indicates that the proposed MRVM still works well under complex scenarios with complicated geometry, realistic non-Lambertian materials and various illuminations.

Table 4: Ablation study on *left*: masking strategies and *right*: masking ratios.

| Method | PSNR↑ | SSIM↑ | LPIPS↓ | #Params(M) |
|---|---|---|---|---|
| NeRFormer | 27.58 | 0.920 | 0.091 | 25.084 |
| RGB mask | 27.95 | 0.925 | 0.080 | 25.934 |
| Feat mask[1] | 28.58 | 0.935 | 0.069 | 25.817 |
| Feat mask[2] | 28.02 | 0.927 | 0.074 | 27.240 |
| NeRFormer+MRVM | **29.25** | **0.942** | **0.060** | 25.151 |

| Mask ratio | PSNR↑ | SSIM↑ | LPIPS↓ |
|---|---|---|---|
| 0.1 | 27.88 | 0.924 | 0.083 |
| 0.25 | 28.54 | 0.930 | 0.076 |
| 0.5 | **29.25** | **0.942** | **0.060** |
| 0.75 | 28.96 | 0.938 | 0.068 |
| 0.9 | 28.02 | 0.927 | 0.080 |

## 4.3 ABLATION STUDY

We execute ablation studies focusing on three aspects as described below.

**Different masking strategies** To validate the influence of different mask-based pretraining strategies, we evaluate with another three masking variants under category-agnostic **ShapeNet-all** setting.

- **RGB mask:** Following MIM, we perform random block-wise masking on reference images and incorporate an additional UNet-like decoder to reconstruct the masked region of pixels.

- **Feat mask[1]:** We take the same masking strategy as described in Section 3.2 but introduce an additional decoder to recover the masked latent feature $\mathbf{h}_i^j$ from the output representation $\mathbf{z}_i^j$.

- **Feat mask[2]:** Similar to our default MRVM but the target network is replaced by a copy of the fine-branch instead of the coarse-branch, with parameters updated via moving average.

The comparisons are shown in Table 4 (left). Although all masking options yield some degree of improvements, our final proposal MRVM achieves the most significant improvement with the minimal additional parameters, demonstrating its superiority over other masking strategies. Please refer to the Appendix for more details about the three variants.

Table 5: Ablation study for **few-shot scenarios** on NeRF Synthetic dataset.

| #views | method | PSNR↑ | SSIM↑ | LPIPS↓ |
|---|---|---|---|---|
| 50-5 | NeuRay | 29.78 | 0.940 | 0.078 |
| | NeuRay+MRVM | **30.88** | **0.948** | **0.060** |
| 20-4 | NeuRay | 25.01 | 0.871 | 0.145 |
| | NeuRay+MRVM | **26.61** | **0.891** | **0.114** |
| 10-3 | NeuRay | 22.19 | 0.809 | 0.208 |
| | NeuRay+MRVM | **24.03** | **0.846** | **0.159** |

**Different masking ratios** We conduct an empirical study on the mask ratio $\eta$ under category-agnostic **ShapeNet-all** setting in Table 4 (right). We separately mask 10%, 25%, 50%, 75% and 90% points along each ray. A relatively-large $\eta$ proves to be more beneficial, as it poses a more challenging pretraining task. It compels the model to develop a comprehensive understanding of the entire 3D scene on a global scale, rather than merely interpolating information from adjacent points. While too large $\eta$ may lead to a too difficult task, it is inappropriate for pretraining to learn sufficient 3D scene prior knowledge.

**Few-shot scenarios** We validate that our MRVM-NeRF could help alleviate the limitation of NeRF's requirement on relatively dense inputs, referred as the **few-shot scenarios** in Table 5. Specifically, we adopt the **per-scene finetuning** setting using NeRF Synthetic dataset. The default configuration in Table 3 uses 100 views for finetuning and renders each image from 8 reference views. For **few-shot scenarios**, we decrease the training views to 50, 20, 10 and reference views to 5, 4, 3 respectively. The results indicate that our MRVM achieves more significant improvements under few-shot scenarios, which implies that the prior knowledge learned through mask-based pretraining holds substantial potential to alleviate the relatively dense inputs required by NeRF.

## 5 CONCLUSION

In this paper, we propose masked ray and view modeling (MRVM), a mask-based pretraining strategy specially designed for generalizable Neural Radiance Field. By enhancing inner correlations among rays and across views, our MRVM shows great efficacy and wide compatibility under various experimental configurations. We hope our work could promote the development of introducing mask-based pretraining into 3D vision research field.

ACKNOWLEDGMENTS

This work was partially supported by the Natural Science Foundation of China under Grant 61931014.

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

## A  MORE EXPERIMENTAL RESULTS

### A.1  RESULTS ON SYNTHETIC DATASETS

**Category-agnostic ShapeNet-all and ShapeNet-unseen settings** The overall numerical results have already been presented in the main paper. The detailed results with a breakdown by categories are provided in Table 6 and Table 7. We provide additional visual results in Figure 10, Figure 11 for **ShapeNet-all** setting and Figure 12, Figure 13 for **ShapeNet-unseen** setting, respectively. We randomly sample 4 object instances for each of the testing categories in ShapeNet dataset and show visual comparisons to PixelNeRF (Yu et al., 2021b) and our baseline NeRFormer.

**Category-specific ShapeNet-car and ShapeNet-chair settings** The quantitative comparisons on PSNR, SSIM and LPIPS are available in the main paper. SRN (Sitzmann et al., 2019), FE-NVS (Guo et al., 2022) and CodeNeRF (Jang & Agapito, 2021) do not provide LPIPS result in their paper. We calculate LPIPS result for PixelNeRF (Yu et al., 2021b) using author-provided checkpoints. More visualizations are shown in Figure 14 and Figure 15. We use view-64 and view-64, 104 as input view(s) for one-shot and two-shot cases. For each scenario we randomly sample 5 object instances, and show visual comparisons to PixelNeRF (Yu et al., 2021b) and our baseline NeRFormer.

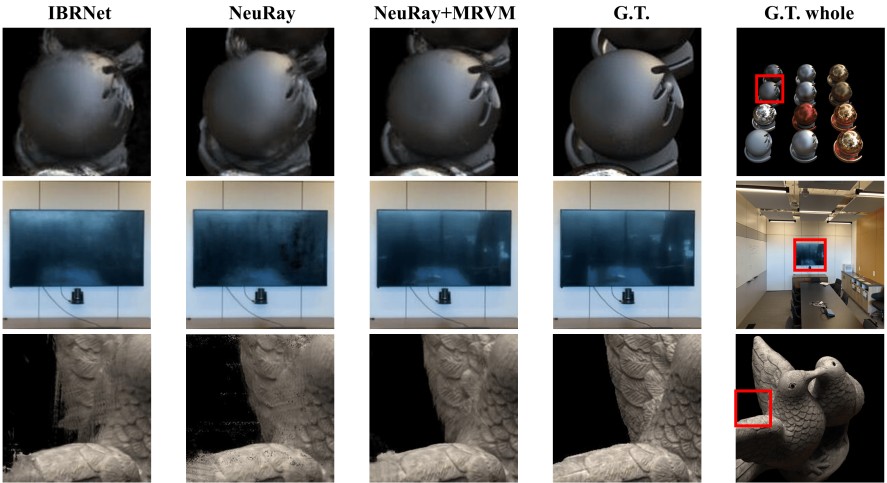

Figure 5: Visualizations for **cross-scene generalization** on NeRF Synthetic (first row), LLFF (middle row) and DTU (last row) datasets.

### A.2  RESULTS ON REALISTIC DATASETS

For real-world **cross-scene generalization** and **per-scene finetuning** settings, as we illustrated in the main paper, we adopt NeuRay (Liu et al., 2022) as baseline and evaluate on three datasets: NeRF Synthetic (Niemeyer et al., 2020), DTU (Jensen et al., 2014) and LLFF (Mildenhall et al., 2019). The quantitative results are presented in Table 3 in the main paper, and more visualizations for **cross-scene generalization** setting and **per-scene finetuning** setting are shown in Figure 5 and Figure 6 respectively.

### A.3  RESULTS ON OTHER BASELINES

We also provide the additional experimental results of adding our proposed masked ray and view modeling (MRVM) on another advanced generalizable NeRF baseline GNT (Wang et al., 2022b), on NeRF Synthetic (Niemeyer et al., 2020) and LLFF (Mildenhall et al., 2019) datasets respectively, and compare with another state-of-the-art method GNT-MOVE (Cong et al., 2023). The default setting for novel-view synthesis is put in Table 8 and the few-shot setting is located in Table 9. We conclude that the proposed masked ray and view modeling consistently benefits under all the cases.

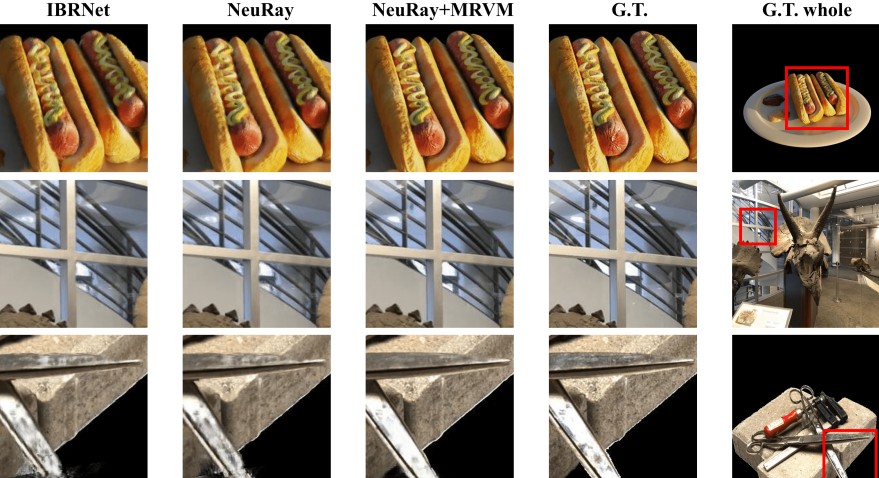

Figure 6: Visualizations for **per-scene finetuning** on NeRF Synthetic (first row), LLFF (middle row) and DTU (last row) datasets.

Table 6: Detailed results of **category-agnostic ShapeNet-all** setting, with a breakdown by categories. This table is an expansion of Table 1 in the main paper.

| Metric | Method | plane | bench | cbnt. | car | chair | disp. | lamp | spkr. | rifle | sofa | table | phone | boat | avg. |
|---|---|---|---|---|---|---|---|---|---|---|---|---|---|---|---|
| PSNR↑ | SRN | 26.62 | 22.20 | 23.42 | 24.40 | 21.85 | 19.07 | 22.17 | 21.04 | 24.95 | 23.65 | 22.45 | 20.87 | 25.86 | 23.28 |
| | PixelNeRF | 29.76 | 26.35 | 27.72 | 27.58 | 23.84 | 24.22 | 28.58 | 24.44 | 30.60 | 26.94 | 25.59 | 27.13 | 29.18 | 26.80 |
| | FE-NVS | 30.15 | 27.01 | 28.77 | 27.74 | 24.13 | 24.13 | 28.19 | 24.85 | 30.23 | 27.32 | 26.18 | 27.25 | 28.91 | 27.08 |
| | SRT | 31.47 | 28.45 | 30.40 | 28.21 | 24.69 | 24.58 | 28.56 | 25.61 | 30.09 | 28.11 | 27.42 | 28.28 | 29.18 | 27.87 |
| | VisionNeRF | 32.34 | 29.15 | 31.01 | 29.51 | 25.41 | 25.77 | 29.41 | 26.09 | 31.83 | 28.89 | 27.96 | 29.21 | 30.31 | 28.76 |
| | NeRFormer | 30.50 | 27.19 | 28.88 | 28.12 | 24.49 | 25.21 | 29.34 | 25.22 | 31.13 | 27.65 | 26.67 | 27.93 | 30.12 | 27.58 |
| | NeRFormer+MRVM | 32.10 | 28.91 | 30.94 | 29.16 | 26.20 | 27.27 | 31.54 | 27.24 | 32.18 | 29.25 | 28.82 | 29.70 | 31.13 | **29.25** |
| SSIM↑ | SRN | 0.901 | 0.837 | 0.831 | 0.897 | 0.814 | 0.744 | 0.801 | 0.779 | 0.913 | 0.851 | 0.828 | 0.811 | 0.898 | 0.849 |
| | PixelNeRF | 0.947 | 0.911 | 0.910 | 0.942 | 0.858 | 0.867 | 0.913 | 0.855 | 0.968 | 0.908 | 0.898 | 0.922 | 0.939 | 0.910 |
| | FE-NVS | 0.957 | 0.930 | 0.925 | 0.948 | 0.877 | 0.871 | 0.916 | 0.869 | 0.970 | 0.920 | 0.914 | 0.926 | 0.941 | 0.920 |
| | SRT | 0.954 | 0.925 | 0.920 | 0.937 | 0.861 | 0.855 | 0.904 | 0.854 | 0.962 | 0.911 | 0.909 | 0.918 | 0.930 | 0.912 |
| | VisionNeRF | 0.965 | 0.944 | 0.937 | 0.958 | 0.892 | 0.891 | 0.925 | 0.877 | 0.974 | 0.930 | 0.929 | 0.936 | 0.950 | 0.933 |
| | NeRFormer | 0.953 | 0.921 | 0.922 | 0.947 | 0.870 | 0.879 | 0.924 | 0.869 | 0.971 | 0.916 | 0.913 | 0.928 | 0.946 | 0.920 |
| | NeRFormer+MRVM | 0.966 | 0.945 | 0.941 | 0.958 | 0.906 | 0.912 | 0.948 | 0.900 | 0.978 | 0.937 | 0.942 | 0.944 | 0.959 | **0.942** |
| LPIPS↓ | SRN | 0.111 | 0.150 | 0.147 | 0.115 | 0.152 | 0.197 | 0.210 | 0.178 | 0.111 | 0.129 | 0.135 | 0.165 | 0.134 | 0.139 |
| | PixelNeRF | 0.084 | 0.116 | 0.105 | 0.095 | 0.146 | 0.129 | 0.114 | 0.141 | 0.066 | 0.116 | 0.098 | 0.097 | 0.111 | 0.108 |
| | FE-NVS | 0.061 | 0.080 | 0.076 | 0.085 | 0.103 | 0.105 | 0.091 | 0.116 | 0.048 | 0.081 | 0.071 | 0.080 | 0.094 | 0.082 |
| | SRT | 0.050 | 0.068 | 0.058 | 0.062 | 0.085 | 0.087 | 0.082 | 0.096 | 0.045 | 0.066 | 0.055 | 0.059 | 0.079 | 0.066 |
| | VisionNeRF | 0.042 | 0.067 | 0.065 | 0.059 | 0.084 | 0.086 | 0.073 | 0.103 | 0.046 | 0.068 | 0.055 | 0.068 | 0.072 | 0.065 |
| | NeRFormer | 0.063 | 0.096 | 0.088 | 0.081 | 0.128 | 0.116 | 0.093 | 0.126 | 0.055 | 0.099 | 0.079 | 0.083 | 0.090 | 0.091 |
| | NeRFormer+MRVM | 0.045 | 0.067 | 0.064 | 0.059 | 0.087 | 0.083 | 0.065 | 0.098 | 0.042 | 0.070 | 0.051 | 0.063 | 0.070 | **0.060** |

# B    MORE IMPLEMENTATION DETAILS

We first provide general configurations that are applicable across all settings, followed by configurations specific to each unique setting.

**General configurations**    For mask-based pretraining, we incorporate $\mathcal{L}_{mrvm}$ as an auxiliary loss. It is optimized together with NeRF's rendering loss not from the beginning, but starting from 10% of the total training iterations until finishing. We also use a warm-up schedule for about 10k iterations which linearly increases the coefficient $\lambda$ from 0 to the final value 0.1. Both of these technical strategies contribute to stabilize the pretraining process. At inference time, we use the VGG network for calculating LPIPS (Zhang et al., 2018) after normalizing pixel values to [-1,1]. We perform ray casting, sampling and volume rendering all in the world coordinate. All the models are implemented using Pytorch (Paszke et al., 2019) framework.

Table 7: Detailed results of **category-agnostic ShapeNet-unseen** setting, with a breakdown by categories. This table is an expansion of Table 1 in the main paper.

| Metric | Method | bench | cbnt. | disp. | lamp | spkr. | rifle | sofa | table | phone | boat | avg. |
|---|---|---|---|---|---|---|---|---|---|---|---|---|
| PSNR↑ | SRN | 18.71 | 17.04 | 15.06 | 19.26 | 17.06 | 23.12 | 18.76 | 17.35 | 15.66 | 24.97 | 18.71 |
|  | PixelNeRF | 23.79 | 22.85 | 18.09 | 22.76 | 21.22 | 23.68 | 24.62 | 21.65 | 21.05 | 26.55 | 22.71 |
|  | FE-NVS | 23.10 | 22.27 | 17.01 | 22.15 | 20.76 | 23.22 | 24.20 | 20.54 | 19.59 | 25.77 | 21.90 |
|  | NeRFormer | 23.64 | 22.21 | 17.77 | 23.20 | 20.60 | 24.11 | 24.58 | 21.05 | 21.24 | 27.32 | 22.54 |
|  | NeRFormer+MRVM | 25.46 | 23.28 | 18.72 | 24.79 | 21.93 | 25.19 | 26.63 | 22.61 | 21.78 | 28.54 | **24.08** |
| SSIM↑ | SRN | 0.702 | 0.626 | 0.577 | 0.685 | 0.633 | 0.875 | 0.702 | 0.617 | 0.635 | 0.875 | 0.684 |
|  | PixelNeRF | 0.863 | 0.814 | 0.687 | 0.818 | 0.778 | 0.899 | 0.866 | 0.798 | 0.801 | 0.896 | 0.825 |
|  | FE-NVS | 0.865 | 0.819 | 0.686 | 0.822 | 0.785 | 0.902 | 0.872 | 0.792 | 0.796 | 0.898 | 0.825 |
|  | NeRFormer | 0.863 | 0.808 | 0.689 | 0.837 | 0.774 | 0.908 | 0.875 | 0.786 | 0.817 | 0.914 | 0.826 |
|  | NeRFormer+MRVM | 0.892 | 0.815 | 0.693 | 0.857 | 0.786 | 0.921 | 0.899 | 0.822 | 0.827 | 0.927 | **0.849** |
| LPIPS↓ | SRN | 0.282 | 0.314 | 0.333 | 0.321 | 0.289 | 0.175 | 0.248 | 0.315 | 0.324 | 0.163 | 0.280 |
|  | PixelNeRF | 0.164 | 0.186 | 0.271 | 0.208 | 0.203 | 0.141 | 0.157 | 0.188 | 0.207 | 0.148 | 0.182 |
|  | FE-NVS | 0.135 | 0.156 | 0.237 | 0.175 | 0.173 | 0.117 | 0.123 | 0.152 | 0.176 | 0.128 | 0.150 |
|  | NeRFormer | 0.141 | 0.175 | 0.243 | 0.181 | 0.185 | 0.109 | 0.127 | 0.177 | 0.182 | 0.101 | 0.159 |
|  | NeRFormer+MRVM | 0.096 | 0.135 | 0.220 | 0.135 | 0.148 | 0.082 | 0.088 | 0.115 | 0.146 | 0.089 | **0.117** |

Table 8: Experimental results of adding our proposed masked ray and view modeling on the baseline of GNT (Wang et al., 2022b) and compare with GNT-MOVE (Cong et al., 2023) on NeRF Synthetic and LLFF datasets.

| Method | Synthetic Object NeRF | | | Real Forward-facing LLFF | | |
|---|---|---|---|---|---|---|
|  | PSNR↑ | SSIM↑ | LPIPS↓ | PSNR↑ | SSIM↑ | LPIPS↓ |
| GNT | 27.29 | 0.937 | 0.056 | 25.86 | 0.867 | 0.116 |
| GNT-MOVE | 27.47 | 0.940 | 0.056 | 26.02 | 0.869 | **0.108** |
| GNT+MRVM | **27.78** | **0.942** | **0.052** | **26.25** | **0.873** | 0.110 |

## B.1 IMPLEMENTATION DETAILS FOR SYNTHETIC DATASETS

Considering the images of synthetic datasets have a blank background, we adopt two techniques following previous works (Yu et al., 2021b; Lin et al., 2022) for better performance. 1) We use bounding box sampling strategy as Yu et al. (2021b) during pretraining, where rays are only sampled within the bounding box of the foreground object. In this way, it avoids the model to learn *too much empty* information at initial training stage. 2) We assign a white background color for those pixels sampled from the background to match the rendering ground truths in ShapeNet dataset.

**Settings** For category-agnostic **ShapeNet-all** setting, we use a batch size of 16, and sample 256 rays per object. We pretrain the model for 400k iterations on 4 GPUs, with a tight bounding box for the first 300k iterations, then we finetune the model without bounding box for 800k iterations. The two-stage training takes about 10 days on GTX-1080Ti.

For category-agnostic **ShapeNet-unseen** setting, we also use a batch size of 16, and sample 256 rays per object. We pretrain for 300k iterations with bounding box on 4 GPUs, and finetune the model for 600k iterations without bounding box, which takes about 8 days on GTX-1080Ti.

For category-specific **ShapeNet-car** and **ShapeNet-chair** settings, we use a batch size of 8, and sample 512 rays per object. We pretrain for 400k iterations on 4 GPUs. For the first 300k iterations, we use 2 input views for the network to encode with a tight bounding box. For the rest of 100k iterations, the bounding box is removed and we randomly choose 1 or 2 view(s) as the input to make the model compatible with both one-shot and two-shot scenarios. We finetune the model for 1 or 2 view(s) respectively on 8 GPUs for 400k iterations. The two-stage training takes about 7 days on GTX-1080Ti.

Table 9: The few-shot experimental results of adding our proposed masked ray and view modeling on the baseline of GNT (Wang et al., 2022b) and compare with GNT-MOVE (Cong et al., 2023) on NeRF Synthetic and LLFF datasets.

| Method | Synthetic Object NeRF | | | | | | Real Forward-facing LLFF | | | | | |
| | 6-shot | | | 12-shot | | | 3-shot | | | 6-shot | | |
| | PSNR↑ | SSIM↑ | LPIPS↓ | PSNR↑ | SSIM↑ | LPIPS↓ | PSNR↑ | SSIM↑ | LPIPS↓ | PSNR↑ | SSIM↑ | LPIPS↓ |
|---|---|---|---|---|---|---|---|---|---|---|---|---|
| GNT | 22.39 | 0.856 | 0.139 | 25.25 | 0.901 | 0.088 | 19.58 | 0.653 | 0.279 | 22.36 | 0.766 | 0.189 |
| GNT-MOVE | 22.53 | **0.871** | **0.116** | 25.85 | **0.915** | **0.074** | 19.71 | 0.666 | 0.270 | 22.53 | 0.774 | 0.184 |
| GNT+MRVM | **23.52** | 0.869 | 0.120 | **26.10** | 0.911 | 0.079 | **20.88** | **0.672** | **0.257** | **23.54** | **0.777** | **0.175** |

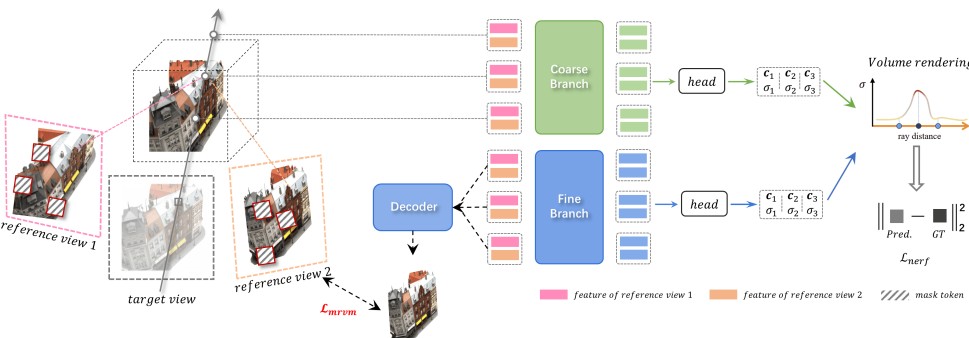

Figure 7: Illustration for mask-based pretraining variant 1 — **RGB mask**. We mask blocks of pixels and try to recover them at pretraining.

## B.2 IMPLEMENTATION DETAILS FOR REALISTIC DATASETS

Following the training protocol in NeuRay (Liu et al., 2022), we first perform cross-scene pretraining across five distinct datasets (Downs et al., 2022; Mildenhall et al., 2019; Flynn et al., 2019; Zhou et al., 2018; Jensen et al., 2014) for 400k iterations. Afterwards, for **cross-scene generalization** setting, we finetune the model on the same five training sets for additional 200k iterations. For **per-scene finetuning** setting, the model is finetuned on each scene respectively in the three testing datasets (Niemeyer et al., 2020; Jensen et al., 2014; Mildenhall et al., 2019) for additional 100k iterations, except for the few-shot scenarios in Table 5 of the main paper where we find only 10k iterations is sufficient for finetuning. When training the generalizable model across multiple datasets, we randomly sample 1 scene from the training sets per iteration. We sample 512 rays for each scene during training. All the training processes are conducted on one V100 GPU, which takes about 5 days for total pretraining and finetuning.

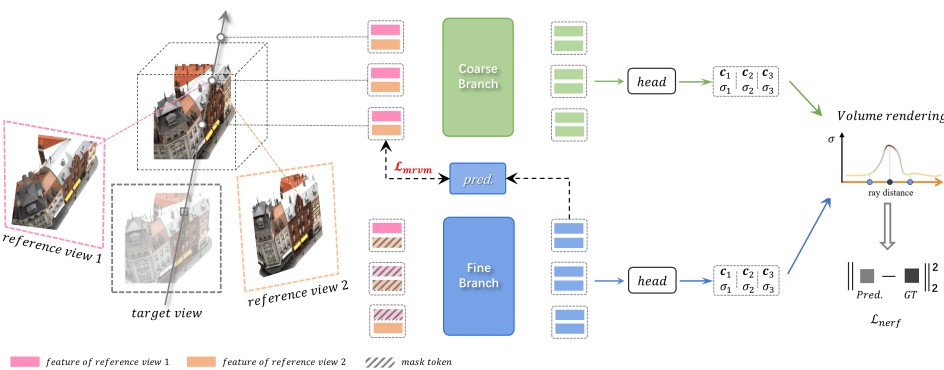

Figure 8: Illustration for mask-based pretraining variant 2 — **Feat mask**[1]**:**. We use the intermediate representation output (boxes in blue) by Fine-Branch to reconstruct the masked feature tokens.

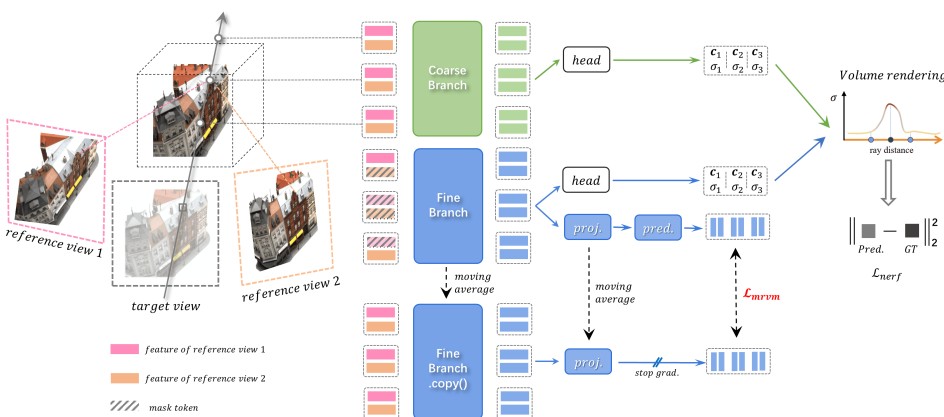

Figure 9: Illustration for mask-based pretraining variant 3 — **Feat mask[2]:**. We make a copy of Fine-Branch as the target branch, in place of Coarse-Branch in the main paper.

### B.3 VARIANTS OF MASK-BASED PRETRAINING OBJECTIVES

As stated in the main paper, we conduct an elaborated ablation study on different mask-based pretraining strategies, which are illustrated in Figure 7, Figure 8 and Figure 9.

- **RGB mask:** As shown in Figure 7, we mask blocks of pixels on input images from reference views. After extracting pyramid features with a 2D CNN, we additionally introduce an UNet-like decoder to recover the masked image pixels based on these features. $\mathcal{L}_{mrvm}$ is the $\mathcal{L}_2$ distance between reconstructed pixels and the ground truth, the constraint is only added to masked regions. We set mask ratio to 50% and patch size to 4 at pretraining.

- **Feat mask[1]:** As illustrated in Figure 8, we perform masking operation on sampled points same as MRVM. Differently, after obtaining intermediate representation $\mathbf{z}_i^j$ from the fine branch, we use it to recover the masked latent feature $\mathbf{h}_i^j$ by a shallow 2-layer MLP. $\mathcal{L}_{mrvm}$ is the $\mathcal{L}_2$ distance between the reconstructed latent feature vector and the unmasked ground truth. We normalize the vector to unit-length before calculating the distance.

- **Feat mask[2]:** The pipeline for this variant is presented in Figure 9. Different from the architecture in the main paper, we don't utilize coarse branch as the target. On the contrary, we make a copy of the fine branch as the target network. With the gradient stopped manually, this branch is updated by moving average of the parameters from the online fine branch. We experimentally find that this option may cause instability at mask-based pretraining stage, making it inappropriate as our final proposal.

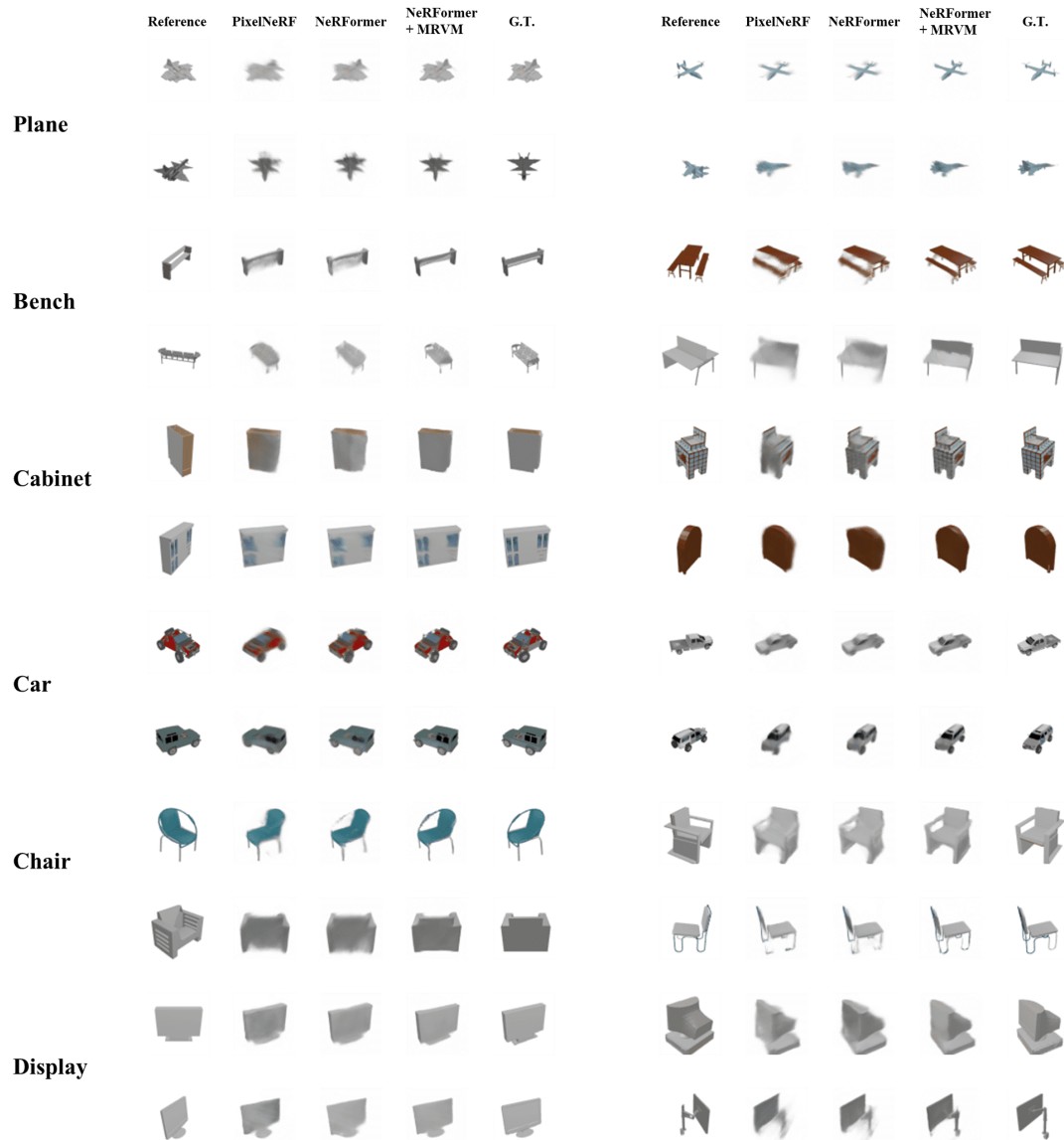

Figure 10: More visualizations for **Category-agnostic ShapeNet-all** setting, Part 1.

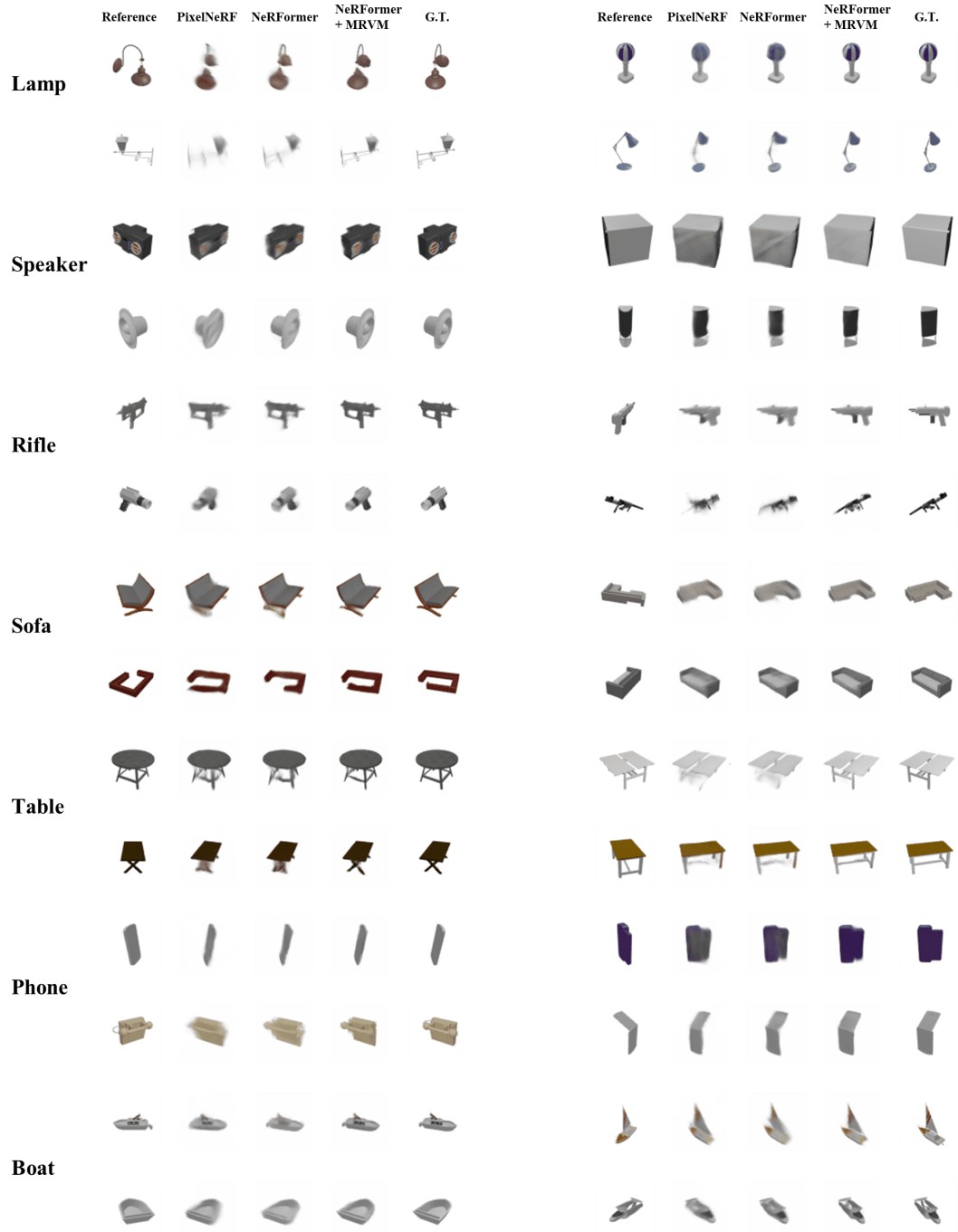

Figure 11: More visualizations for **Category-agnostic ShapeNet-all** setting, Part 2.

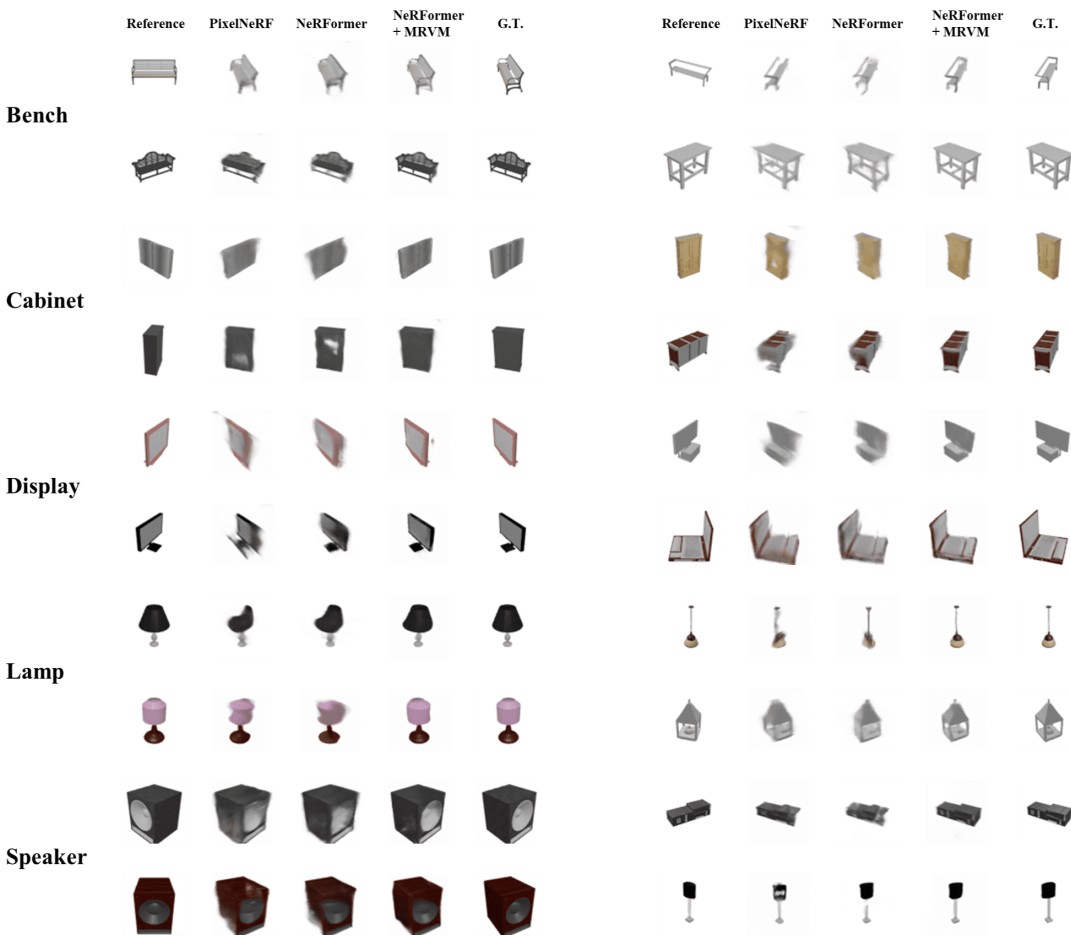

Figure 12: More visualizations for **Category-agnostic ShapeNet-unseen** setting, Part 1.

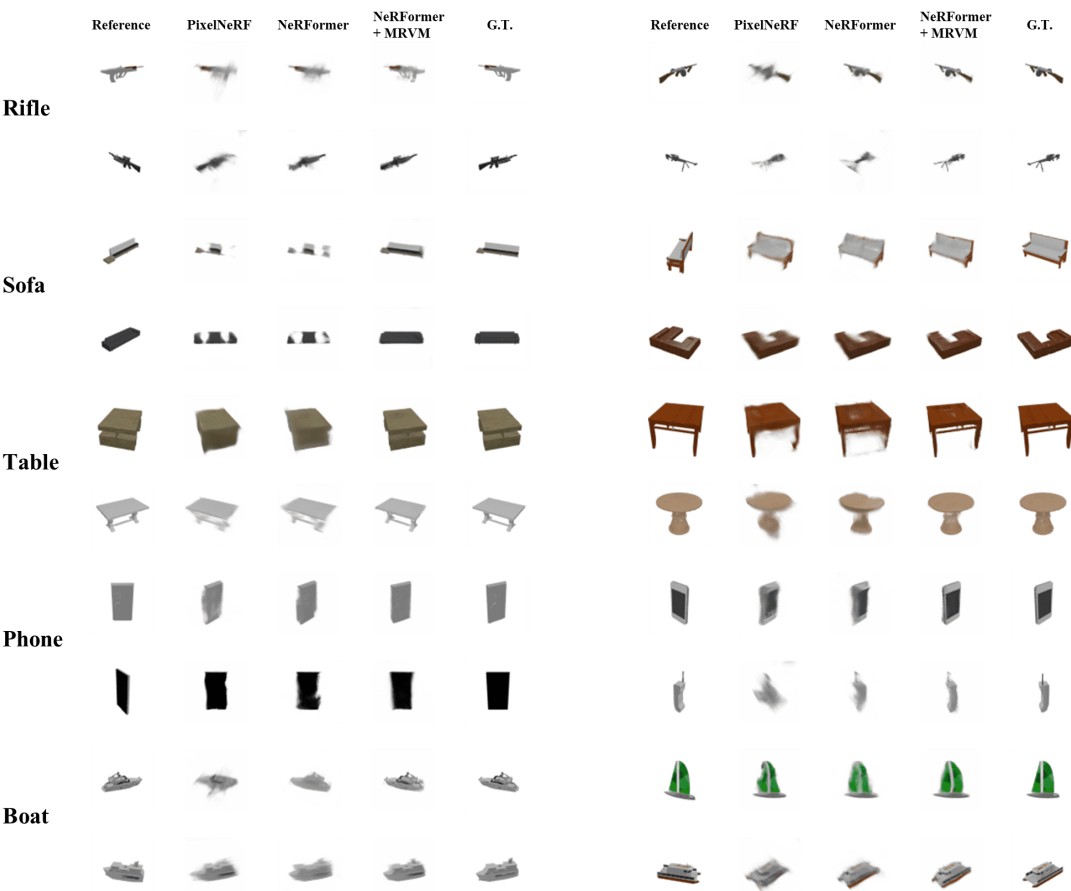

Figure 13: More visualizations for **Category-agnostic ShapeNet-unseen** setting, Part 2.

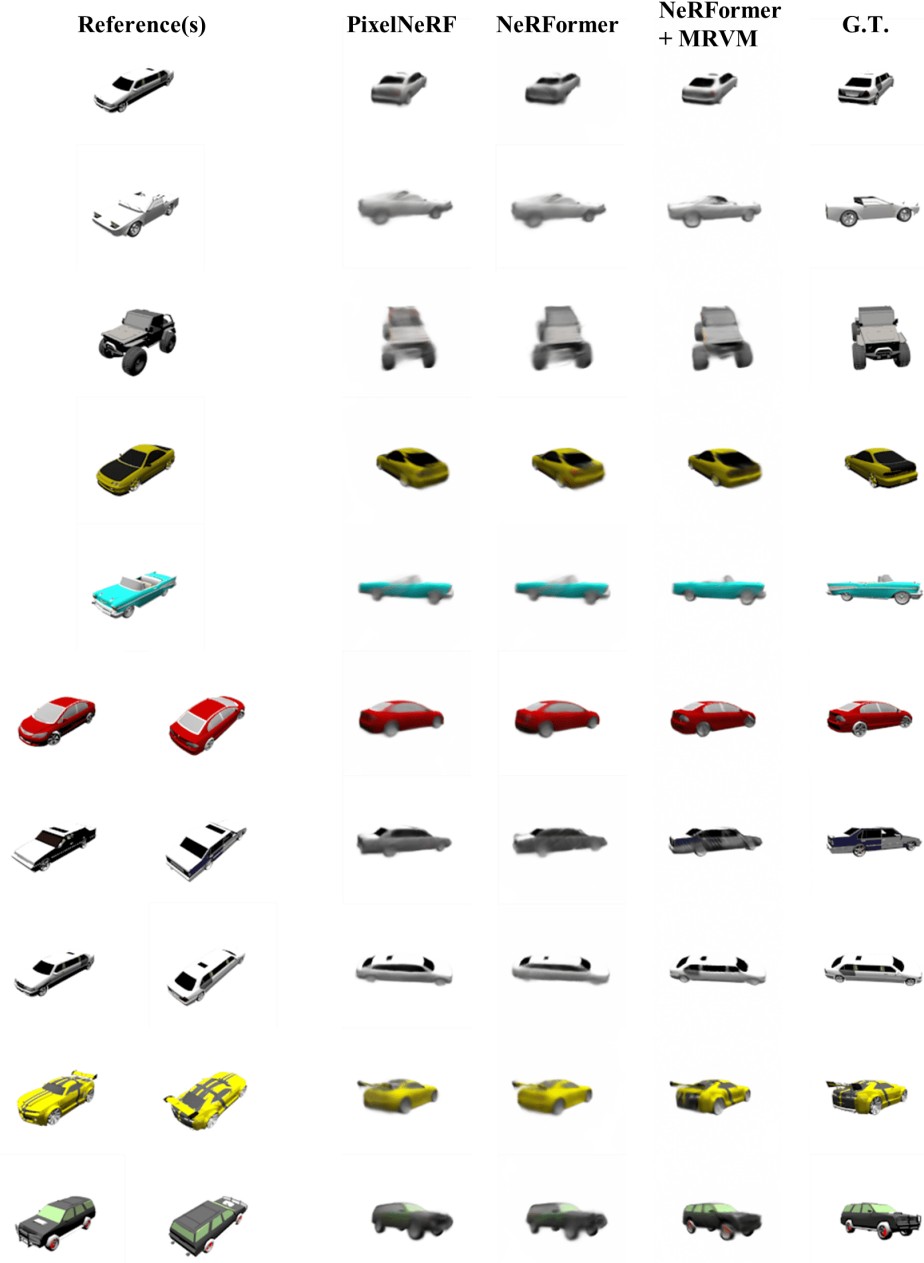

Figure 14: More visualizations for **Category-specific ShapeNet-car** setting.

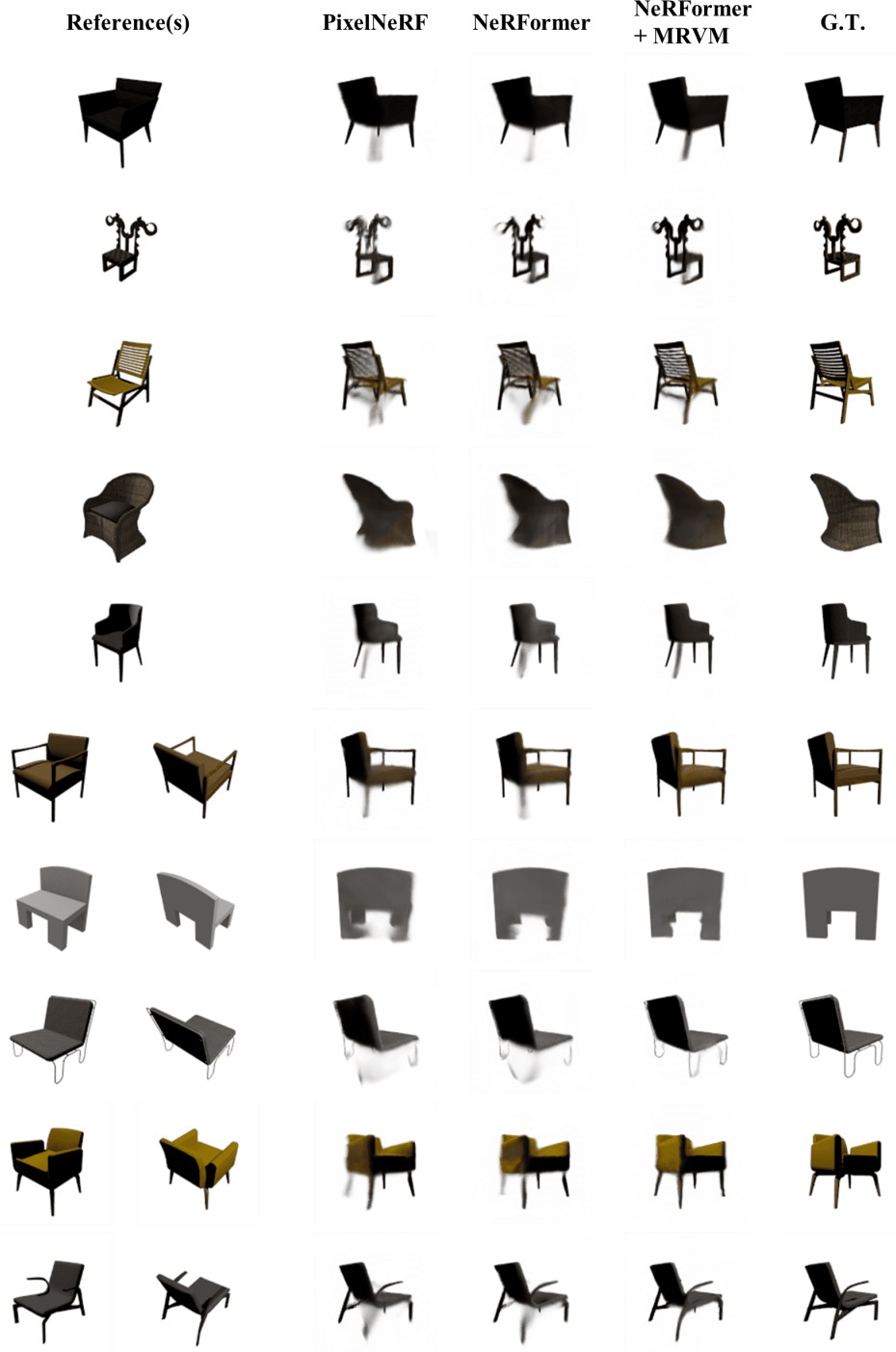

Figure 15: More visualizations for **Category-specific ShapeNet-chair** setting.

