# OpenReview forum: "Mask-Based Modeling for Neural Radiance Fields"
_ICLR.cc/2024/Conference — ICLR 2024 spotlight_

### Official Review · Reviewer_GXVJ · 2023-10-30

**Soundness:** 4 excellent
**Presentation:** 3 good
**Contribution:** 3 good
**Rating:** 8
**Confidence:** 4

**Summary:**

This paper aims to tackle the Generalizable NeRF task, which represents multiple scenes using a single NeRF model. The authors introduce masked ray and view modeling (MRVM) to demonstrate that 3D implicit representation learning can be improved by mask-based modeling. In practice, the masking mechanism in MRVM carries out a ray level to enhance the information interaction along each ray and also a view level to promote the message-passing across different reference views. The numerical experiments and visualizations show the efficiency of the proposed method on several NeRF image reconstruction datasets.

**Strengths:**

•	The paper is easy to follow, with well-written paragraphs and good section/figure/table organization.

•	The proposed masked ray and view modeling is sound. Experiments demonstrate the effectiveness and superiority of the proposed method.

**Weaknesses:**

* Concerning the different masking strategies, i.e., RGB mask and two feature masks, it is curious whether ray-level or view-level has the more considerable performance gain or whether each masking strategy prefers a different level of masking.

* The MRVM merely carried out upon one MLP-based NeRF model, i.e., NeuRay (Liu et al., 2022), and one Transformer-based NeRF model, i.e., NeRFormer (Reizenstein et al., 2021). It would be great if more SOTA MLP/Transformer-based models could be integrated with MRVM for comparison.

**Questions:**

In short, this paper gives a well-written method description for readers. Some minor concerns are that the experiments for deeper discussion and comparison, as shown in [weakness], could be conducted.

---

> ### Author Response · Authors · 2023-11-21
> **Official Response to Reviewer GXVJ**
>
> Dear Reviewer GXVJ,
>
> Thank you for your comments. Please see below our response to your concerns.
>
> **Weakness 1**: Do different masking strategies prefer different levels of masking?
>
> **Answer 1**:  In our experiments, we find that the combination of using ray-level as well as view-level masking operation brings the most significant improvements, no matter for our final proposal or for the other three masking variants (i.e., RGB mask and two feature masks). We reckon that masking along the two dimensions together will better exploit the inner correlations among points and across views, in which case the 3D scene prior knowledge will be learnt more adequately. We already search for the optimal masking level (i.e., masking ratio) for our final proposal as well as the three masking variants in the ablation study, and report the best for each in *Table 4*, which shows the superiority of our final proposal.
>
> **Weakness 2**: It would be great if more generalizable NeRF models could be integrated with MRVM for comparison.
>
> **Answer 2**:  We add our mask-based modeling strategy to another well-known generalizable NeRF baseline GNT [1], under the cross-scene generalization setting, using LLFF and NeRF Synthetic as evaluation sets. The quantitative comparisons are shown below:
>
> | Models | LLFF (PSNR$\uparrow$) | LLFF (SSIM$\uparrow$) | LLFF (LPIPS$\downarrow$) | Synthetic (PSNR$\uparrow$) | Synthetic (SSIM$\uparrow$) | Synthetic (LPIPS$\downarrow$) |
> | :----- | :-----------------: | :---------------: | :----------------: | :-----------------: |:-----------------:|:-----------------:|
> | GNT    | 25.86 | 0.867 | 0.116 | 27.29 | 0.937 | 0.056 |
> | GNT+MRVM   | **26.25** | **0.873** | **0.110** | **27.78** | **0.942** | **0.052** |
>
> Our proposed MRVM proves to be effective in improving the generalizability for GNT as well.
>
> [1]  Mukund Varma T, Peihao Wang, Xuxi Chen, Tianlong Chen, Subhashini Venugopalan, Zhangyang Wang. Is attention all that nerf needs?

---

> ### Comment · Reviewer_GXVJ · 2023-11-23
> **Official Comment by Reviewer GXVJ**
>
> The reviewer thanks the authors for their responses. After reading all the other reviewers’ comments, I would like to keep my rating to endorse this paper for acceptance.

---

### Official Review · Reviewer_y3Pz · 2023-10-30

**Soundness:** 3 good
**Presentation:** 3 good
**Contribution:** 2 fair
**Rating:** 6
**Confidence:** 4

**Summary:**

In this paper, the authors introduce masked-based learning and propose a self-supervised pretraining for generalizable NeRF. Specifically, they randomly mask feature token $h_i^j$ along cast rays and across refrence views, then introduce a BYOL-like module to reconstruct missing information in $\bar z$ space. In this way, the proposed method can utilize the correlations among point-to-point and across view-to-view and learn a 3D scene prior knowledge. In addition, extensive experiments are conducted to validate the effectiveness of the proposed method.

**Strengths:**

+ The proposed method is effective. It boosts the performance of generalizable NeRF on different datasets largely. Meanwhile, the proposed method is easy to implement due to its simplicity.
+ This paper is written well and easy to understand, although there are some missing details in the main paper.

**Weaknesses:**

- In the experiments, NeuRay is adopted as the MLP-based network. But, to my knowledge, the MLP-based NeRF processes each sampled 3D point  independetly. It means that it is not possible to utilize the correlations among point-to-point and across view-to-view and learn a 3D scene prior knowledge. Are there some architecture modifications for NeuRay?
- To validate the influence of different masking strategies, the authors conduct experiments with three masking variants. There is missing one variant: taking the same masking strategy as described in Section 3.2 but minimize the $\mathcal L_2$ distance between $z_i^f$ and $z_i^c$. Note that normalizing the vector to unit-length before calculating the distance. This additional experiment can validate the effectiveness of the BYOL-like module.

**Questions:**

- As shown in Weaknesses, there are some missing details about the architecture of the used backbones. It is better to provide these details in the paper.

---

> ### Author Response · Authors · 2023-11-21
> **Official Response to Reviewer y3Pz**
>
> Dear Reviewer y3Pz,
>
> Thank you for your comments. Please see below our response to your concerns.
>
> **Weakness 1**:  Are there any architecture modifications for NeuRay?
>
> **Answer 1**:  We do not make any architecture modifications for NeuRay. Naturally, there exist inherent correlations among the sampled points and across the reference views for generalizable NeRFs, regardless of different network architectures. Our mask-based pretraining target is to well-exploit such correlations for improving NeRF's generalizability. Even for the MLP-based NeRFs, such correlations, as the 3D scene prior knowledge, can still be learnt through our proposed MRVM and encapsulated in the implicit MLP networks as well.  That's because MLP networks also have some kind of inner correlations among different input signals. Our  experiments demonstrate the applicability of our proposed MRVM for both MLP-based and Transformer-based backbones.
>
> **Weakness 2**: An additional ablation experiment is needed to validate the effectiveness of the BYOL-like module.
>
> **Answer 2**:  Thanks for your valuable suggestion, we add the ablation as the reviewer suggested and refer as **NeRFormer+MRVM (w/o BYOL)** in the table below:
>
> | Method    | PSNR$\uparrow$ | SSIM$\uparrow$ | LPIPS$\downarrow$ |
> | --------- | ---- | ---- | ----- |
> | NeRFormer | 27.58 | 0.920 | 0.091 |
> | NeRFormer+MRVM (original)| **29.25** | **0.942** |**0.060**|
> | NeRFormer+MRVM (w/o BYOL)| 28.09 | 0.924 |0.086|
>
> It validates the effectiveness of the BYOL-like module in our final proposal. Intuitively, as many previous contrastive-learning methods [1,2] pointed out,  the usage of projector and predictor avoid the model collapsing into a trivial solution, that is, the two branches learn similar but meaningless representations, which impairs the effectiveness of pretraining.
>
> [1] Jean-Bastien Grill, Florian Strub, Florent Altche, Corentin Tallec, Pierre Richemond, Elena Buchatskaya, Carl Doersch, Bernardo Avila Pires, Zhaohan Guo, Mohammad Gheshlaghi Azar, et al. Bootstrap your own latent-a new approach to self-supervised learning.
>
> [2] Ting Chen, Simon Kornblith, Mohammad Norouzi, Geoffrey Hinton   A Simple Framework for Contrastive Learning of Visual Representations.
>
> **Question 1**:  It is better to provide more details about the architecture of the used backbones in the paper.
>
> **Answer 3**:  Thanks for your valuable suggestion, we'll provide more details about the architectures of the used backbones in our revision to make them more clear. We'll put them in the revised appendix.

---

> > ### Comment · Reviewer_y3Pz · 2023-11-22
> >
> > I have read all reviews and authors’ rebuttal. The authors resolve my concerns. This paper is good to accept, but it is not beyond my expectation. Thus, I will keep my rate.

---

### Official Review · Reviewer_fmpu · 2023-11-04

**Soundness:** 3 good
**Presentation:** 3 good
**Contribution:** 3 good
**Rating:** 8
**Confidence:** 4

**Summary:**

The authors propose MRVM-NeRF, a framework that leverages an additional masked training objective to improve rendering and generalization ability. The method uses multi-view input of a scene to perform novel view synthesis. The masking is executed along cast rays and across reference views, along with an online fine branch supervised by the coarse stage. Results indicate that the method achieves SOTA generalization performance on simple (ShapeNet) and complex (NeRF Synthetic, LLFF, and DTU) datasets. The masked-based pretraining also improves finetuning performance. The authors also conduct experiments on few-shot finetuning on the NeRF synthetic dataset and achieve better performance.

**Strengths:**

- The paper is easy to read and well-presented.
- All the design choices are well-motivated.
- Further quantitative results indicate performance improvements, across multiple datasets and settings (generalization, scene-specific tuning, and few-shot finetuning).

**Weaknesses:**

There are no major weaknesses, in my opinion, design choices are quite well motivated, and quantitative results do indicate improvements. I think it might be interesting to investigate the few-shot results in more detail (i.e. show that the pretraining objective now enables tasks that were previously not looked at by generalizable nerfs).
- For example, since the model is trained on masked inputs - it must additionally be able to perform quite well on cross-scene generalization in a few-shot setting (w/o any finetuning). Even in the case of finetuning, [1] presents some results on few-shot finetuning with as little as 3-6 images on LLFF, and 6-12 on NeRF Synthetic. It might be worth comparing against the same.

[1] Enhancing NeRF akin to Enhancing LLMs: Generalizable NeRF Transformer with Mixture-of-View-Experts

**Questions:**

- In the case of online training, where supervision comes from the coarse branch, I wanted to clarify, do you use the same randomly sampled input points to both coarse and fine branches, in addition to the (masked) importance sampled inputs used by the fine branch?
- I wonder, do the results depend on the choice of masking? Have any other strategies than random masking been tried out (for example removing geometrically consistent regions across all the views)?

---

> ### Author Response · Authors · 2023-11-21
> **Official Response to Reviewer fmpu**
>
> Dear Reviewer fmpu,
>
> Thank you for your comments. Please see below our response to your concerns.
>
> **Weakness 1**:  It might be interesting to investigate the few-shot results in more detail, like the way GNT and GNT-MOVE does.
>
> **Answer 1**:  Thanks for your valuable suggestion. We add our proposed Masked Ray and View Modeling to a well-known generalizable NeRF baseline GNT [1], and compare under the same few-shot settings as the reviewer mentioned:
>
> | Models | LLFF (3-shot)       | LLFF (6-shot)     | Synthetic (6-shot) | Synthetic (12-shot) |
> | ------ | ------------------- | ----------------- | ------------------ | ------------------- |
> | GNT    | 19.58  0.653  0.279 | 22.36 0.766 0.189 | 22.39 0.856 0.139  | 25.25 0.901 0.088   |
> | GNT+MRVM   | **20.88**  **0.672**  **0.257** | **23.54** **0.777** **0.175** | **23.52** **0.869** **0.120** | **26.10** **0.911** **0.079** |
>
> The numerical numbers are PSNR, SSIM and LPIPS from left to right respectively. It shows our proposed MRVＭ still benefits under the few-shot settings.
>
> [1] Mukund Varma T, Peihao Wang, Xuxi Chen, Tianlong Chen, Subhashini Venugopalan, Zhangyang Wang. Is attention all that nerf needs?
>
> **Question 1**:  A clarification for the details of the way in performing masking operation.
>
> **Answer 2**:  As the reviewer understands, we use the same randomly sampled input points to both coarse and fine branches, in addition to the additional importance sampled inputs used by the fine branch. However, we perform randomly masking operation to all the points input into the fine branch, not just to the additional importance sampled ones. Please refer to Section 3.2 for more details.
>
> **Question 2**:  Have any other masking strategies rather than random masking been tried out?
>
> **Answer 3**:  This paper proposes a general framework for incorporating mask-based pretraining strategy into the NeRF community. We only use random masking to prove its effectiveness. Thanks for the reviewer's suggestions, we believe more specially-designed masking choices are worth exploring in the future works.

---

> > ### Comment · Reviewer_fmpu · 2023-11-23
> >
> > The authors present some additional results that in the few-shot setting that are particularly appealing. I would highly encourage the authors to include these in the final version of the manuscript.
> >
> > Overall, I do endorse this paper for acceptance.

---

### Official Review · Reviewer_chDn · 2023-11-04

**Soundness:** 3 good
**Presentation:** 3 good
**Contribution:** 3 good
**Rating:** 8
**Confidence:** 4

**Summary:**

This paper investigates a Mask-based pertaining strategy called masked ray and view modeling (MRVM) for a generalizable Neural Radiance Field. It is the first attempt to incorporate mask-based pretraining into the NeRF field. To fit the NeRF field, a simple yet efficient self-supervised pretraining objective is proposed. Abundant experiments demonstrate its benefits for different architectures and data categories. I wonder if the training code and models be released.

**Strengths:**

- Mask-based self-supervised pretraining has been demonstrated to benefit wide NLP and CV tasks. This work firstly attempts to introduce mask-based pretraining into NeRF field.

- The designed masking strategy and objectiveness are suitable for the NeRF field. Obvious and consistent improvements can be obtained in various settings.

- The discussion and analysis of "prior NeRFs lack an explicit inductive bias from other views and points" and "distinct scale learning of two branches” are good, which could benefit the architecture design.


- Abundant experiments have been constructed in different settings to demonstrate the effectiveness of the proposed method.
Experiments on different network architectures (i.e. MLP and transformer) demonstrate its generalization to different models.
Experiments on cross-scene and per-scene fine-tuning settings indicate the benefits of the mask-based pertaining to a wide range of scenes, including complicated geometry, and realistic non-Lambertian materials.
Furthermore, the setting of the few-shot scenario reveals its significant improvements on the few-views setting (10-3 views).

**Weaknesses:**

- If I understand correctly when applying to the proposed method, an additional fine branch would be added. I wonder if this will cause double parameters and inference costs.

- As to the method part, the terminology is not easy to follow. What is the projector, predictor?
Besides, the parameter updating procedure is not very clear.  For example, it is claimed that “the parameters of coarse-projector are updated by moving average from the fine-projector”, what is the updating procedure for other parts?

- The explanation of different masking strategies in the ablation study of the main paper is not easy to understand. I see the details are well illustrated in the supplementary material. I recommend claiming these illustrations in the main paper as well.

**Questions:**

The parameter updating strategy and discussion on the cost of the additional fine branch are necessary.

---

> ### Author Response · Authors · 2023-11-21
> **Official Response to Reviewer chDn**
>
> Dear Reviewer chDn,
>
> Thank you for your comments. We'll release the code and models after acceptance. Please see below our response to your concerns.
>
> **Weakness 1**:  Whether the proposed method will cause double parameters and inference costs?
>
> **Answer 1**:   Most of the Neural Radiance Field works adopt coarse-to-fine sampling strategy, using two independent network modules handling sampled points at coarse and fine stage respectively. We reuse the two network module branches as target and online branch, aiming to introduce the fewest additional parameters. Therefore, the only additional parameters introduced by our MRVM are the *light-weight projector and predictor*, which brings negligible additional computational overhead (only +0.2% parameters compared to the baseline model). Please refer to *Table 4* for more details. Moreover, since the projector and predictor are discarded at inference time, the inference speed is consistent with the original baseline methods.
>
> **Weakness 2**:  The terminology in method section is not easy to follow. Besides, what is the parameter updating procedure for other network parts apart from the projector?
>
> **Answer 2**:  We use the terminology 'projector' and 'predictor' following the previous work BYOL [1].  The *projector* represents the neural network projecting the intermediate feature to another latent space and the *predictor* represents the neural network predicting target representations from online ones. As shown in Figure 1, since the gradient is stopped manually for the target projector, it could only be updated using moving average from the online counterpart. Noting that we still adopt vanilla NeRF's rendering loss $L_{nerf}$ for supervision during our mask-based pretraining stage, as shown in Equation 10. Therefore, the rest of other neural network modules are updated using gradient-decent algorithm as before. Thanks for your valuable suggestion, we'll make the statements more clearly in our revision.
>
> [1] Jean-Bastien Grill, Florian Strub, Florent Altche, Corentin Tallec, Pierre Richemond, Elena Buchatskaya, Carl Doersch, Bernardo Avila Pires, Zhaohan Guo, Mohammad Gheshlaghi Azar, et al. Bootstrap your own latent-a new approach to self-supervised learning.
>
> **Weakness 3**: Suggestions on illustrating the masking strategies in ablation study with more details in the main paper.
>
> **Answer 3**:  Thanks for your valuable suggestion. Due to the page limits, we are only able to describe the three masking variants briefly in the main paper, we'll try to make them more precise in our revision.

---

> > ### Comment · Reviewer_chDn · 2023-11-22
> > **Thanks for the authors' responses**
> >
> > I appreciate the authors’ responses to my feedback. My concerns have been addressed. I endorse this paper for acceptance.

---

### Official Review · Reviewer_Jc3W · 2023-11-04

**Soundness:** 4 excellent
**Presentation:** 4 excellent
**Contribution:** 3 good
**Rating:** 6
**Confidence:** 4

**Summary:**

The paper presents several significant improvements to the standard generalizable-NeRF framework, in which a NeRF is trained on a set of scenes and is used at inference on a novel scene without training. It successfully incorporates some recent advances from self-supervised representation learning, which include the use of masked prediction, exponential moving average learning, using transformer-based backbones. The main contribution is in the way these techniques are introduced into the NeRF framework, with the particular choice of the student network acting in the 'fine' sampling branch with random masking and the teacher network acting on the 'coarse' branch. Results verify the method, with significant and consistent improvements in the different metrics across the board.

**Strengths:**

1] The proposed method shows very good performance compared to the baselines, both significant and consistent, across all experimentation. This is apparent visually and quantitatively, though an extensive set of experiments.
2] The idea of introducing masking into the training scheme is well motivated, as a means for improving generalization to new scenes. It extends the standard pipeline quite naturally and does not impact inference time complexity. The decision to mask at two levels, within the ray and across views is interesting.
3] Incorporating student-teacher learning within NeRF is also quite a natural thing to do. It gives further regularization that is likely responsible for the improved generalization.
4] Paper is well written and organized, especially the method and experiments that explain and demonstrate the advantages very clearly.

**Weaknesses:**

1] The two main additions - of EMA and masking are tested together. There is a lack in understanding how dependent they are. Firstly, if one could be used without the other. And if so, secondly, what are the individual contributions.
2] Sampling is not well specified. The extra 'fine' samples - whether they are they taken, as in the classical NeRF, according to an initial estimation of density from the coarse sample. And in this respect - shouldn't the masking prediction be focused at the more important (close-to-surface) high density locations, rather than at the coarse sampling?
3] Impact on training time is not clear. Even though training time and procedures are very clearly presented in the appendix, I am missing what is the addition in comparison to the baseline, and which components are responsible for it?
4] Clarity [minor]: (i) I would suggest improving Figure 1 to include some of the notations (g, h, etc') ; (ii) What are the blue dashed arrows in Figure 2 supposed to represent?

**Questions:**

0] Please relate to above 'weaknesses'
In addition:
1] Have you tried comparing to per-scene NeRFs?
2] Could you provide an ablation showing the contribution of each of the components? In particular - applying only one of EMA / Masking?
5] How sensitive is the choice of \lambda, the weight of the mrvm-loss?

---

> ### Author Response · Authors · 2023-11-21
> **Official Response to Reviewer Jc3W**
>
> Dear Reviewer Jc3W,
>
> Thank you for your comments. Please see below our response to your concerns.
>
> **Weakness 1**: Can EMA and masking be used without the other rather than tested together?
>
> **Answer 1**:  We'd like to firstly explain our approach of using EMA and masking together. As we mentioned in Section 3.2, apart from the vanilla rendering loss, we use an additional feature-alignment loss $L_{mrvm}$ in latent space following the successful formulation introduced by BYOL [1]. This requires the gradient to be manually stopped at the target (coarse) branch, as we illustrated in Figure 1. Therefore the projector in target branch can not be updated with gradient decent algorithm, that's why we use EMA to update its parameters using Equation 5. If only using masking without EMA, as we understand, is to make a copy of the parameters of the projector from online branch to target branch every iteration. We find this will make the mask-based pretraining stage instable based on our experiments $-$ the loss term $L_{mrvm}$ fluctuates and is not easy to converge.
>
> [1] Jean-Bastien Grill, Florian Strub, Florent Altche, Corentin Tallec, Pierre Richemond, Elena Buchatskaya, Carl Doersch, Bernardo Avila Pires, Zhaohan Guo, Mohammad Gheshlaghi Azar, et al. Bootstrap your own latent-a new approach to self-supervised learning.
>
> **Weakness 2**: Shouldn't the masking prediction be focused at the more important (close-to-surface) high density locations, rather than at the coarse sampling?
>
> **Answer 2**:  As the reviewer understands, the extra fine samples are taken, as in the classical NeRF, according to an initial estimation of density from the coarse samples. We agree that the close-to-surface points sampled at fine stage are more important, but we do not treat them as prediction target due to the fact that : 1) We can not predict the latent embeddings of these points directly from the coarse branch since they are newly sampled at fine stage and do not exist at the coarse stage. That makes it impractical to treat coarse branch as online and fine branch as target. 2)  We do perform an ablation study of making a copy of the overall fine-branch to serve as target and update via moving average, dubbed as **Feat mask^2^** in Section 4.3. However we find this variant, with more additional parameters, exhibits inferior performance compared to our final proposal. Please refer to *Table 4*  for quantitative comparisons and refer to the **Discussion** part in **Section 3.2** for more detailed analysis.
>
> **Weakness 3**: Impact on training time is not clear.
>
> **Answer 3**:  The generalizable NeRF adopts the training strategy of cross-scene pretraining followed by per-scene finetuning. It is firstly pretrained across various scenes. After pretraining, the model can either be tested directly on a novel scene or further finetuned on that specific scene. Our proposed Masked Ray and View Modeling serves as an auxiliary task when performing cross-scene pretraining, which requires an additional finetuning stage without masking to maintain the consistency between training and testing. Overall it takes about 30% additional training time compared to the baselines.
>
> **Weakness 4**: Some minor suggestions.
>
> **Answer 4**:   The blue dashed arrows represent the message-passing along sampled points on the ray. Thanks for your valuable suggestion, we'll include such notations in our revision.
>
> **Question 1**: Have you tried comparing to per-scene NeRFs?
>
> **Answer 5**:  The **per-scene finetuning** results as shown in Table 3 have already be comparable to several per-scene NeRF methods, and it shrinks the gaps compared to SOTA performance on per-scene NeRFs.
>
> **Question 2**: Could you provide an ablation showing the contribution of applying only one of EMA / Masking?
>
> **Answer 6**:  The ablation for different mask-based pretraining strategies are presented in *Table 4*. We find only applying masking will make the mask-based pretraining instable. As mentioned in **Answer 1**, if copying the parameters from online projector to target ones every iteration, we observe fluctuation for the loss term $L_{mrvm}$, making it not easy to converge in our experiments.
>
> We do not try to only apply EMA without masking since we propose a mask-then-predict pretraining task, simply abandoning masking operation is irrelevant to our proposal. Besides, the rest of the network parameters except the target projector are updated using normal gradient-decent algorithm, with no need for the EMA parameter updating strategy.
>
> **Question 3**: How sensitive is the choice of $\lambda$, the weight of the mrvm-loss?
>
> **Answer 7**:  Experimentally we find $\lambda$ suitable between 0.1~0.5.

---

### Official Review · Reviewer_P3FF · 2023-11-06

**Soundness:** 3 good
**Presentation:** 3 good
**Contribution:** 3 good
**Rating:** 8
**Confidence:** 4

**Summary:**

This paper proposes a novel masked ray and view modeling for generalizable NeRF. With mask-based pretraining, the model can learn 3D scene prior knowledge which is useful for reconstructing a high-quality new scene from limited reference views. Experiments show that the MRVM-NeRF achieves state-of-the-art novel view synthesis with limited views and cross-scene generalization.

**Strengths:**

1. The idea is novel and interesting. The authors introduce mask-based pretraining into NeRF, which provides 3D scene prior knowledge for  NeRF generalization.
2. The experimental results are impressive. The MRVM-NeRF achieves superior quantitative and qualitative results compared with other methods.
3. This paper is well-written and easy to understand.

**Weaknesses:**

1. Why do the authors use masked-based modeling to learn high-level global information? What are the advantages and necessity of using masked-based modeling?
2. The authors should add the masked modeling design to more generalizable NeRF models to demonstrate its wide applicability. In addition, the authors should provide qualitative and quantitative comparisons with SOTA generalizable NeRF models, such as FreeNeRF[1] and SparseNeRF[2].

[1] Yang et al. FreeNeRF: Improving Few-shot Neural Rendering with Free Frequency Regularization. In CVPR, 2023.
[2] Wang et al. SparseNeRF: Distilling Depth Ranking for Few-shot Novel View Synthesis. In ICCV, 2023.

**Questions:**

Can this method only handle object-centric and facing-forward scenes? In addition, can the authors provide results on the tank-and-temple datasets, where the scenes have a wider range of views and complex backgrounds.

---

> ### Author Response · Authors · 2023-11-21
> **Official Response to Reviewer P3FF**
>
> Dear Reviewer P3FF,
>
> Thank you for your comments. Please see below our response to your concerns.
>
> **Weakness 1**: Why using masked-based modeling to learn high-level global information and what are the advantages and necessities?
>
> **Answer 1**:  In our **Abstract**, **Introduction** and **Method 3.1** Section, we have emphasized the motivation and benefits of using mask-based modeling for improving the generalizability of Neural Radiance Field. A mask-then-predict pretraining objective has been widely applied in MLM [1] and MIM [2] field for learning a high-level global representation. As in NeRFs, we reckon that the learnt 3D scene global information, which contains the correlations among sampled points and across reference views, endowing the model with better capacity to effectively generalize to novel scenes with limited observations.
>
> [1]  Jacob Devlin, Ming-Wei Chang, Kenton Lee, and Kristina Toutanova. Bert: Pre-training of deep bidirectional transformers for language understanding.
>
> [2]  Kaiming He, Xinlei Chen, Saining Xie, Yanghao Li, Piotr Dollar, and Ross Girshick. Masked autoencoders are scalable vision learners.
>
> **Weakness 2**: The authors should add the masked modeling design to more generalizable NeRF models to demonstrate its wide applicability.
>
> **Answer 2**:  We add our mask-based modeling strategy to another well-known generalizable NeRF baseline GNT [3], under the cross-scene generalization setting, using LLFF and NeRF Synthetic as evaluation sets. The quantitative comparisons are shown below:
>
> | Models | LLFF (PSNR$\uparrow$) | LLFF (SSIM$\uparrow$) | LLFF (LPIPS$\downarrow$) | Synthetic (PSNR$\uparrow$) | Synthetic (SSIM$\uparrow$) | Synthetic (LPIPS$\downarrow$) |
> | :----- | :-----------------: | :---------------: | :----------------: | :-----------------: |:-----------------:|:-----------------:|
> | GNT    | 25.86 | 0.867 | 0.116 | 27.29 | 0.937 | 0.056 |
> | GNT+MRVM   | **26.25** | **0.873** | **0.110** | **27.78** | **0.942** | **0.052** |
>
> Our proposed MRVM proves to be effective in improving the generalizability for GNT as well.
>
> [3]  Mukund Varma T, Peihao Wang, Xuxi Chen, Tianlong Chen, Subhashini Venugopalan, Zhangyang Wang. Is attention all that nerf needs?
>
> Besides, we'd like to clarify that *FreeNeRF and SparseNeRF as the reviewer mentioned are **not** generalizable NeRF methods* which may not be compared to our proposed approach targeting on generalizable NeRFs directly. These two methods focus on solving the few-shot reconstruction problem especially designed for *a single particular scene*, while generalizable NeRF methods use a shared network to *represent various different scenes*.
>
> **Question 1**:  Can this method only handle object-centric and facing-forward scenes?
>
> **Answer 3**:  We currently perform experiments on the object-centric and forward-facing scenes because the previous generalizable NeRF baselines are mainly evaluated on these two kinds of scenes. For more complicated scenes like in-the-wild unbounded 360 degree scenes as in the tank-and-temple dataset, or the large-scale complex indoor scenes, most SOTA methods in the NeRF community still learn an independent NeRF per scene for satisfying rendering quality. However, we see no obstacles that may hinder our proposed method in applying for other types of scenes, and we'll leave these as a future work.

---

> ### Comment · Reviewer_P3FF · 2023-11-22
> **Official comment by Reviewer P3FF**
>
> Thanks for the author’s detailed responses. I have read the response and other reviewers’ feedback. The authors address my doubt and so I raise my score to accept.

---

### Official Review · Reviewer_s7zj · 2023-11-09

**Soundness:** 3 good
**Presentation:** 3 good
**Contribution:** 3 good
**Rating:** 6
**Confidence:** 4

**Summary:**

The paper under review introduces a self-supervised pretraining strategy termed Masked Ray and View Modeling for Generalizable Neural Radiance Fields (MRVM-NeRF). The proposed method uses a dual-branch network: an unmasked coarse branch serving as the target and a masked fine branch acting as the online network. By training the masked fine branch to predict the unmasked coarse branch, the authors show that the model can learn better representations and generalize better to unseen scenes. The author claims that this work is the first to adapt the concept of masked self-supervised learning to the domain of generalizable NeRF.

**Strengths:**

1. This method is able to achieve better performance than the baseline methods on various datasets and evaluation settings (category agnostic/specific, generalization/fine-tuning, few-shot, etc.).

2. This method does not require additional supervision data or priors.

3. This method can be applied to various types of generalizable NeRFs (e.g., NeRFormer and NeuRay) as a plug-in module.

**Weaknesses:**

1. As this method adds an additional branch to the baseline network, it is not as computationally efficient as the baseline methods. The computational overhead is not well discussed in the paper.

2. Baseline generalizable NeRFs are already doing "pretraining".  I think this paper's contribution is to **improve** the pretraining by adding a self-supervised masked prediction task to the baseline methods. The author may need to clarify this point in the paper.

**Questions:**

1. Can the authors provide more details on the computational efficiency of MRVM-NeRF?

2. To predict coarse latent features, we take the masked latent feature as input to the prediction network (Equation 4). Can the $Pred^f$ function take additional inputs besides the latent feature? E.g., geometrical

---

> ### Author Response · Authors · 2023-11-21
> **Official Response to Reviewer s7zj**
>
> Dear Reviewer s7zj:
>
> Thank you for your comments. Please see below our response to your concerns.
>
> **Weakness 1**: The computational overhead is not well discussed in the paper.
>
> **Answer 1**:  Most of the Neural Radiance Field works adopt coarse-to-fine sampling strategy, using two independent network modules handling sampled points at coarse and fine stage respectively. We reuse the two network module branches as target and online branch, aiming to introduce the fewest additional parameters. Therefore, the only additional parameters introduced by our MRVM are the *light-weight projector and predictor*, which brings negligible additional computational overhead (only +0.2% parameters compared to the baseline model). Please refer to *Table 4* for more details. Moreover, since the projector and predictor are discarded at inference time, the inference speed is consistent with the original baseline methods.
>
> **Weakness 2**:  The author may need to clarify the role of mask-based pretraining more clearly in the paper.
>
> **Answer 2**:  Yes we agree with the reviewer's comment. Thanks for the valuable suggestion and we will clarify this point in our revision.
>
> **Question 1**: Can the authors provide more details on the computational efficiency of MRVM-NeRF?
>
> **Answer 3**:  Please refer to **Answer 1**.
>
> **Question 2**: Can the $Pred^f$ function take additional inputs besides the latent feature like geometrical information?
>
> **Answer 4**:  Theoretically, $Pred^f$ can take additional geometrical inputs like position **$x$** and view direction **$d$** by modifying the predictor network structure. It's worth exploring whether this operation will bring greater performance gain in the future work. Actually, the geometrical information has already been introduced at the beginning of the generalizable NeRF framework. In Section 3.1, the input latent embedding $h_i^j$ has already contained the information of position $x$ and direction $d$ $-$ in the way that $x$ and $d$ are projected into a latent embedding after positional encoding, which are then added to $f_j^j$ , $c_i^j$ to produce input vector $h_i^j$. We apologize for the lack of these details and will clarify them in our revision.

---

> > ### Comment · Reviewer_s7zj · 2023-12-05
> > **Thank authors for the response**
> >
> > Thank the authors for the response. Some of my questions are addressed in the rebuttal, and the authors promised to add the clarification of mask-based pretriaining in the revised paper (but not explicitly explained in the rebuttal). So I will keep my original rating.

---

### Meta-Review · Area_Chair_LYip · 2023-12-11

**Metareview:**

**Summary**


This paper presents MRVM-NeRF, a self-supervised retraining framework for generalizable NeRF.  Specifically, when learning NeRF on multi-view images, a BYOL-style self-distillation objective is used to predict randomly masked feature tokens along casting rays. In this way, the proposed method learns 3D scene prior, which shows effectiveness on several NeRF benchmarks.


**Strengths**

1. The proposed method is the first work introducing mask-based retraining in the context of NeRF, which fills up the missing piece towards representation learning for neural fields.

2. The proposed method is general and can be plugged-in any generalizable NeRF models.

3. The experimental results are significant and superior compared to other methods.

4. The reuse of coarse and fine networks as the teacher and student models in pre-training is well-motivated.



**Weaknesses**

1. Additional comparison on computational overhead at both training and inference time should be discussed.

2. It lacks some necessary clarifications and discussions including the choice of EMA, masking strategies and confusing terminology.

**Justification For Why Not Higher Score:**

Some details of the paper are missing and need some clarification. Also, the paper did not discuss how the pre-trained representations can be used for other downstream tasks instead of reconstruction.

**Justification For Why Not Lower Score:**

The paper received uniformly positive feedback, and all reviewers agreed on the significance of the proposed pre-training methods for NeRF.

---

### Decision · Program_Chairs · 2024-01-16

Accept (spotlight)